# Multiple parameters shape the 3D chromatin structure of single nuclei at the doc locus in *Drosophila*

Markus Götz [1,2], Olivier Messina [1], Sergio Espinola [1], Jean-Bernard Fiche[1] & Marcelo Nollmann [1] ✉

The spatial organization of chromatin at the scale of topologically associating domains (TADs) and below displays large cell-to-cell variations. Up until now, how this heterogeneity in chromatin conformation is shaped by chromatin condensation, TAD insulation, and transcription has remained mostly elusive. Here, we used Hi-M, a multiplexed DNA-FISH imaging technique providing developmental timing and transcriptional status, to show that the emergence of TADs at the ensemble level partially segregates the conformational space explored by single nuclei during the early development of *Drosophila* embryos. Surprisingly, a substantial fraction of nuclei display strong insulation even before TADs emerge. Moreover, active transcription within a TAD leads to minor changes to the local inter- and intra-TAD chromatin conformation in single nuclei and only weakly affects insulation to the neighboring TAD. Overall, our results indicate that multiple parameters contribute to shaping the chromatin architecture of single nuclei at the TAD scale.

Chromatin in interphase nuclei is organized at multiple levels, including chromosome territories, A/B compartments, and topologically associating domains (TADs)[1]. TADs, first observed in ensemble-averaged Hi-C contact maps[2–4], are sub-megabase genomic regions of preferred contacts and three-dimensional (3D) proximity. In mammalian cells, loop extrusion by the cohesin/CTCF system contributes to TAD formation[5]. TADs often encapsulate *cis*-regulatory elements (CREs), thereby facilitating interactions between enhancers and promoters that are critical for transcriptional regulation[6–11]. At the same time, TAD borders may also restrict interactions between CREs located in neighboring TADs[4,12,13]. The precise interplay between TADs, enhancer-promoter (EP) contacts, and transcriptional activation is currently under intense study[14]. Cell type-specific EP contacts have been observed[15–17] and direct visualization of EP interactions suggests that sustained physical proximity is necessary for transcription[18]. On the other hand, other studies suggested that enhancer action may not require loop formation between enhancers and promoters[19–21]. TADs arise during the early stages of development, concomitantly with the activation of zygotic gene expression[22,23]; however, emergence of

TADs seems to be independent of transcription itself in most organisms[22,24–26]. Remarkably, chromatin structure at the TAD scale is cell-type independent[27,28] and does not change upon transcriptional activation[28] during early *Drosophila* development. Thus, it is still unclear whether the formation of TADs contribute to transcriptional regulation and what is the role played by single-cell heterogeneity.

Several lines of evidence clearly established that chromosome organization is highly heterogeneous between cells. On the one hand, single-cell Hi-C[29] revealed that the conformation of individual TADs and loops varies substantially during interphase[30,31], along the cell cycle[32], and during early development[33]. On the other hand, imaging-based technologies showed that physical chromatin contacts within and between TADs are rare[34,35], and display large cell-to-cell variability[34–37]. Finally, a high degree of heterogeneity in chromatin organization at the TAD scale was also present in polymer model simulations[38–40]. Overall, these studies suggest that TADs may exist in the ensemble but not in single cells.

This hypothesis was recently challenged using super-resolution microscopy. Several studies observed that chromosomes folded into

[1]Centre de Biologie Structurale, Univ Montpellier, CNRS UMR 5048, INSERM U1054, Montpellier, France. [2]Present address: PicoQuant GmbH, Rudower Chaussee 29, 12489 Berlin, Germany. ✉e-mail: marcelo.nollmann@cbs.cnrs.fr

'nano-compartments' possibly representing 'TAD-like domains'[41,42]. The condensation of chromatin within nano-domains tends to correlate with epigenetic state[41]. Borders between nano-compartments appear to be permissible, with a substantial overlap between neighboring regions[41,43,44]. Consistently, borders between nano-compartments detected in single cells do not necessarily align with ensemble TAD boundaries[37]. Thus, TAD-like domains exist but display different structural properties between different single cells. How these structural properties relate to single-cell chromatin structures and to transcriptional regulation is currently unclear. This is in part because of limitations in ensemble sequencing-based chromosome conformation capture (3C) methods that cannot simultaneously capture chromosome structure and transcriptional status in single cells, and to limitations in conventional fluorescence in situ hybridization (FISH) techniques that can only visualize a very limited number of genomic loci at once.

Here, we investigate chromatin organization in single-nuclei before and after the emergence of TADs during Drosophila embryogenesis. For this, we resorted to Hi-M, a microscopy-based chromosome conformation capture method that simultaneously detects the 3D position of multiple genomic loci and their transcriptional status in single cells. We find a large heterogeneity in single-nucleus chromatin conformations, independent of the presence of a TAD border in the population-average. Remarkably, despite this heterogeneity, chromatin structures of nuclei from different developmental stages segregate in a high-dimensional conformation space. This segregation cannot be assigned to any specific structural parameter. Notably, the single-nucleus chromatin organization of transcriptionally active and inactive nuclei are indistinguishable, both between TADs and inside a TAD. Therefore, chromatin organization is not predictive for the transcriptional state at the single-nucleus level. Finally, the spatial separation of genomic regions from two neighboring TADs is similar for active and inactive nuclei, indicating that physical encapsulation of gene regulatory elements may not be a strict necessity for the spatio-temporal control of transcription during early Drosophila embryogenesis.

## Results

### Quantification of single-nucleus chromosome organization heterogeneity during early Drosophila embryogenesis

After fertilization, the fruit fly embryo undergoes thirteen rapid and synchronous nuclear division cycles (nc1 to nc14). During these cycles, nuclei continuously decrease their volumes due to the increase in the number of nuclei and to their migration to a reduced region close to the periphery of the embryo[45]. Transcription by the zygote is initiated in two waves: the minor wave between nc9-nc13, and the major wave at nc14[46–48]. This last stage coincides with the emergence of ensemble TADs[22]. Thus, early Drosophila embryonic development represents an ideal model system to investigate whether heterogeneity in chromosome organization changes with nuclear condensation, with the onset of transcription, and/or with the emergence of TADs.

To address this question, we monitored changes in 3D chromatin organization at the TAD scale between nc11 and nc14 using HiM, a single nucleus multiplexed imaging method that allows for the single nucleus reconstruction of chromatin architecture in whole-mount embryos (Fig. 1a)[36,49]. We applied Hi-M to a locus displaying two TADs (TAD1 and doc-TAD) (Fig. 1b) annotated using Hi-C data[22]. The doc-TAD contains three developmental genes (doc1, doc2, and doc3)[50] that are specifically activated at nc14 in the dorsal ectoderm[28] (Fig. 1b). The doc genes encode functionally redundant T-box transcription factors essential for the development of the amnioserosa and cardiogenesis[51], and display similar, well-defined, cross-shaped expression patterns in the blastoderm embryo (nc11 to nc14). Nuclei and barcodes were segmented, localized, and drift-corrected as indicated previously (Fig. S1a-b[28,49]), and ensemble Hi-M pairwise distance (ePWD) maps for

embryos at nc11/12 and nc14 were constructed by kernel density estimation of the full pairwise distance distributions (Figs. 1b, and S1c-d, Methods). The most notable difference between nc11/12 and nc14 resided in an overall decrease in distances within TAD1 (Fig. 1c, yellow box), and an overall condensation of doc-TAD in nc14 nuclei (Fig. S1e). Notably, a considerable number of pairwise distances between TAD1 and doc-TAD barcodes (Fig. 1c, pink box) increased in nc14, despite the overall compaction of the locus in this nuclear cycle. These results are consistent with the emergence of ensemble TADs at nc14[22], but do not shed light into the possible changes in structural heterogeneity between nuclear cycles.

To characterize variability in chromatin organization during early Drosophila development, we estimated the similarity in 3D conformation between different single nuclei. For this, we calculated the correlation between PWD maps of single-nuclei (snPWD) following the method of Conte et al.[40]. We performed pair-correlation analysis for embryos at nc11/12 and at nc14, before and after the emergence of ensemble TADs. In this analysis, a pair correlation of 1 corresponds to identical chromatin organization (apart from a possible constant scaling factor) while a pair correlation of 0 corresponds to no correlation between the PWD between two nuclei. Notably, the PWD maps of individual nuclei displayed a large degree of heterogeneity, with few nuclei exhibiting a PWD map similar to that of the ensemble (Fig. 1d, e, top panels). For both developmental stages, the distribution of pair corDOIrelations were broad and centered around zero, indicating that the chromatin conformations of most nuclei were different from each other. Nevertheless, the distribution was skewed towards positive values (Fig. 1d, e, black line indicates median of the distribution), and the integral of the curve was lower for negative than for positive pair-correlations (0.6 versus 0.4 for both Fig. 1d, e), indicating that only a small proportion of nuclei displayed similar chromatin conformations. The overall lack of similarity between individual nuclei was observed before and after ensemble TADs emerged (nc11/12 versus nc14, Fig. 1d, e). We note that the lack of similarity between single-nuclei and ePWD maps is perhaps not surprising, as the average of a multi-parametric, widespread distribution tends not to represent any single individual[52]. Overall, these analyses indicate that chromosome structure heterogeneity is not only present in cultured cells[34,35,38] but is also common during early Drosophila development, despite the presence of robust mechanisms to control and coordinate gene expression and to synchronize the cell cycle of different nuclei.

### Presence of an ensemble TAD border partially segregates the conformational space explored by single nuclei

Our previous analyses indicate that nc14 and nc11/12 nuclei display distinct ensemble average conformations despite their highly heterogeneous chromatin organizations (Fig. 1c–e). However, these analyses do not reveal the extent to which the conformational spaces explored by single nc14 and nc11/12 nuclei overlap, and different scenarios can be envisaged: (1) chromatin conformations between two conditions are distinct, leading to their segregation in conformational space; (2) Conformations are partially shared between conditions, leading to an overlap of the occupied conformation space; or (3) Conformations are largely the same between two conditions (Fig. 1f). To address this issue, we turned to analysis techniques that do not rely on averages but rather on exploring chromatin conformations of single nuclei. The total number of independent dimensions needed to describe the accessible chromatin conformational space can be estimated by the number of degrees of freedom. Even for a small number of barcodes (20), this results in a 54-dimensional space which is inaccessible using conventional plotting methods. Thus, we turned to Uniform Manifold Approximation and Projection (UMAP), an unsupervised, nonlinear dimension-reduction approach previously used to represent single-cell chromatin conformations[53]. In our implementation, we used UMAP to embed snPWD maps in a 2D space (Methods).

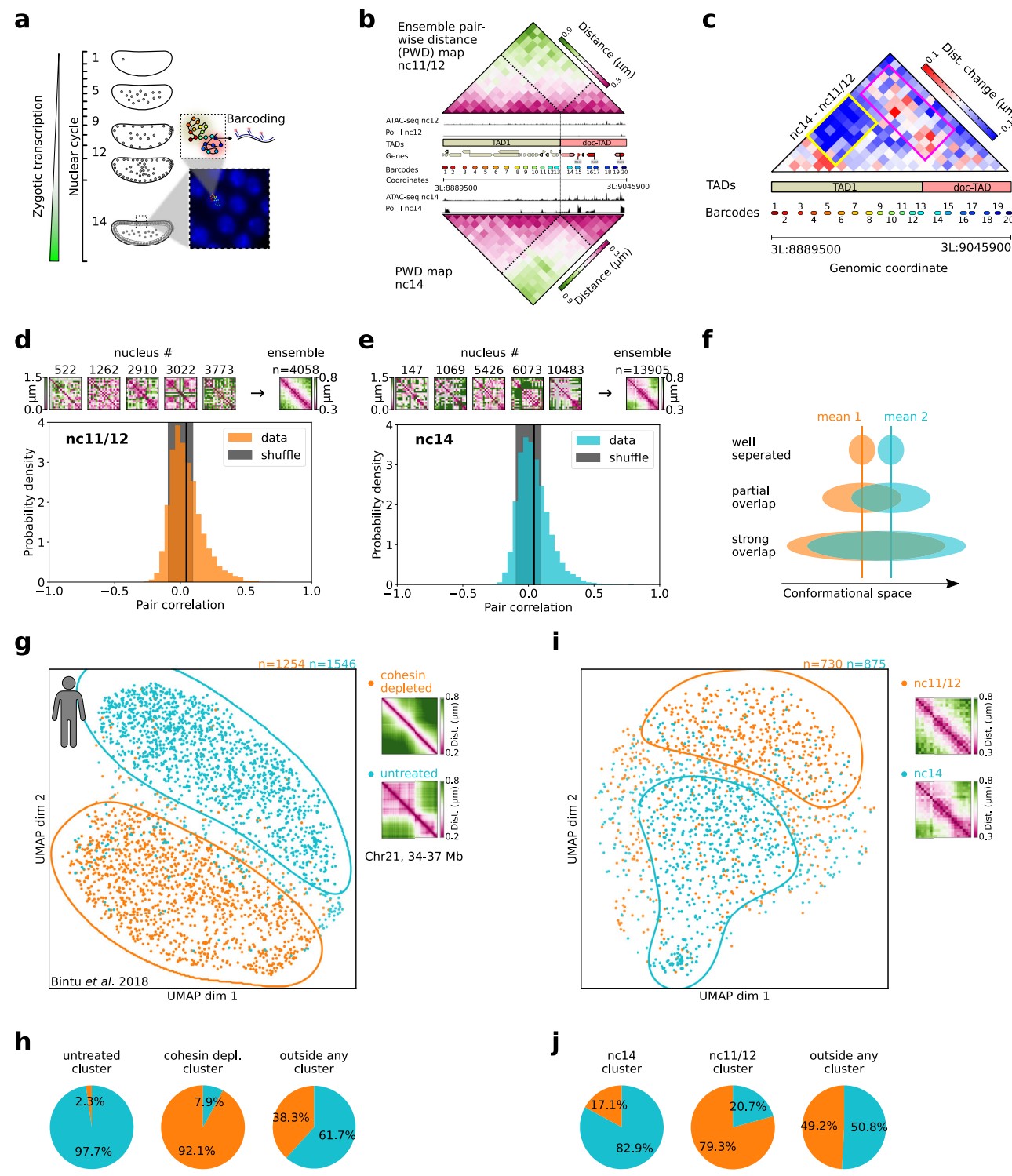

To validate our approach, we applied it to published single-cell data from cultured human cancer cells (HCT116)[37]. Untreated cells displayed two clearly visible TADs that faded considerably in cohesin-depleted cells (Fig. 1g, right panels). The UMAP embedding of single-cell PWD maps showed two clearly separated populations highlighted by orange and cyan contours (Fig. 1g, left panel), delimiting a smaller region where single cells were mixed. We note that a small number of cohesin-depleted single nuclei (~3%) localized to the region occupied by untreated cells, and vice versa (~8%, Fig. 1h). We observed similar segregation patterns for subsets containing 20 barcodes spanning either a strong or a weak TAD border (Fig. S1f).

Thus, UMAP embedding of snPWD maps shows that removal of TAD borders by cohesin depletion dramatically changes the chromatin structure of most single cells at this locus.

To study whether this change also occurred during the natural emergence of TADs during embryonic development, we applied UMAP embedding to snPWD maps at nc11/12 and nc14. Remarkably, we observed that most single nuclei segregated into two distinct populations corresponding to the different nuclear cycles (Fig. 1i). As for human cultured cells, single nuclei occupied extended regions of the UMAP (Fig. 1i), consistent with a large degree of heterogeneity in chromosome structure (Fig. 1d, e). A small number of nc11/12 single

**Fig. 1 | The conformational space explored by chromatin conformations of single nuclei evolves during development. a** Scheme of the nuclear positions during the early *Drosophila melanogaster* development and oligopaint-FISH labeling and barcoding strategy. Inset shows nc14 DAPI-stained nuclei. **b** Extended genomic region (chr3L:8.8895–9.0459 Mb, dm6) around the *doc* locus. Hi-M ensemble pairwise distance maps of nc11/12 and nc14 are shown on top and bottom, respectively. Tracks of regions with accessible chromatin and with RNA polymerase are shown for both developmental stages. TAD calls from ref. 22, and position of genes and barcodes are indicated. Dashed lines in Hi-M maps represent the positions of ensemble TADs from ref. 22. Number of nuclei for the ePWD maps: $n = 4085$ (nc11/12), $n = 13905$ (nc14). **c** Change in ensemble pairwise distance maps between nc11/12 and nc14. PWD larger in nc14 than in nc11/12 are shown in red. Pink box highlights the region between TAD1 and doc-TAD, and yellow box highlights a subTAD within TAD1. **d** Single-nucleus similarity of the chromatin organization during nc11/12. Top: Five sn-PWD maps and ensemble PWD map. Distances (in μm) are color coded according to the colorbars. Bottom: Histogram of the pair correlation for all pairs of nuclei. Number of nuclei: $n = 730$, number of pairs = 6720.

Black vertical line indicates the mean pair correlation. Gray area shows the mean +/− standard deviation of the pair correlation for randomly shuffled PWD maps. **e** Similar to d, but for nc14. $n = 1413$, number of pairs = 14475. **f** Scheme of the chromatin conformational space in three hypothetical scenarios. **g** Left: UMAP of the sn chromatin organization in human HCT116 cells in untreated (cyan) and cohesin-depleted (orange) conditions. Contours with solid lines highlight regions with a high density of cells from one condition. Data taken from ref. 37, locus chr21:34.6Mb-37.1 Mb. Number of nuclei: $n = 1254$ (cohesin-depleted), $n = 1546$ (untreated). Right: Ensemble-average PWD maps for both conditions. **h** Quantification of percentage of nuclei in untreated, and cohesin-depleted clusters (see panel **g**), and outside of either of them. **i** Similar to **g**, but for data from this study (nuclei of intact *Drosophila* embryos), nc11/12 (orange) and nc14 (cyan). Number of nuclei: $n = 730$ (nc11/12), $n = 875$ (nc14). Number of replicates for panels **a**–**e**, **i** = 2, number of embryos for nc11/12 = 7, for nc14 = 9. **j** Quantification of percentage of nuclei in the nc14, and the nc11/12 clusters (see panel **i**), and outside either of them. Source data for panels **b**, **d**, **e**, **g**, and **i** are provided as a Source Data file.

nuclei (~17%) was found in the nc14 cluster, while ~20% of nc14 single nuclei localized to the nc11/12 cluster (Fig. 1j). Segregation of single-nuclei in two populations was not sensitive to variations of the UMAP hyperparameters (Fig. S1g). Therefore, we conclude that despite a large degree of heterogeneity, the chromatin structures of single nuclei change considerably during the early cycles of *Drosophila* embryogenesis and occupy distinct conformational spaces. The mechanisms driving the segregation of single chromatin conformations in HCT116 cells and in *Drosophila* nuclei are likely to be different, as CTCF/cohesin does not seem to be the main actor in the formation of TADs in *Drosophila*[54]. All in all, we conclude that developmental timing partially explains the distribution of single nuclei in the UMAP space.

## TAD condensation is not critical to distinguish between single nuclei conformations

Next, we investigated whether other parameters additionally contributed to this distribution. First, we tested the degree to which TAD condensation influenced the segregation of single nuclei within the UMAP conformational space. For this, we measured the radius of gyration ($R_g$) of the two TADs in single nuclei, as a proxy for TAD volume. Adjacent TAD volumes were highly heterogeneous and uncorrelated (Pearson coefficient ~0.1), but decreased from nc11/12 to nc14 (Fig. 2a). This latter observation raises the possibility that changes in TAD volumes may contribute to the segregation of the two distinct UMAP populations. To explore this hypothesis, we color-coded each nucleus in the UMAP by the radius of gyration of the doc-TAD (Fig. 2b). Volumes in nuclei within the nc14 population tended to exhibit smaller volumes ($R_g < 0.27$ μm), while the opposite was observed in the nc11/12 population ($R_g > 0.27$ μm). However, nuclei in both regions displayed extended and compact TAD volumes and no clear trend of the radius of gyration along either UMAP dimension was discernible. Thus, TAD volume only partially contributes to the global separation of single nuclei between nuclear cycle clusters within the UMAP space.

## Single nuclei displaying insulated TADs are common before the emergence of ensemble TADs

As *Drosophila* TADs arise in nc14[22], we reasoned that TAD insulation may determine how single nuclei occupy the UMAP space. To test this hypothesis, we first calculated the ensemble and single nuclei insulation scores (IS) of nc14 nuclei (Fig. 2c, top panel) following the method developed by Crane et al.[55]. Low insulation scores represent regions with strong borders (see Methods). As expected, the median IS profile for nc14 nuclei showed a dip at the barcode located at the border between TAD1 and doc-TAD (hereafter *border barcode*). The corresponding dip in the median IS profile for nc11/12 was instead less pronounced (Fig. 2d, top panel), consistent with the emergence of the

ensemble TAD border at nc14 in this locus (Fig. 1b, c). We explored the variability of this border by stacking the single-nuclei IS profiles, sorted by their IS score at the border barcode (Fig. 2c, d bottom panels). We quantified the proportion of nuclei displaying a strong insulation between TADs by using a single-nucleus IS cutoff of 3.5. Notably, at nc14 only a small proportion of single nuclei (~14% with IS < 3.5) displayed insulation at the border barcode (i.e., the ensemble TAD border), while many nuclei exhibited low insulation scores at other genomic locations. Similar results were obtained for other IS cutoffs (Fig. S2a). Thus, we conclude that for nc14 embryos the border barcode is not always insulated but rather represents the region displaying the most preferred insulation, consistent with results in human cultured cells[37].

As ensemble TADs emerge at nc14, we wondered whether and to what extent single nuclei in previous nuclear cycles were already insulated at the border barcode. As expected, the ensemble IS at the border barcode was higher in nc11/12 than in nc14 nuclei (Fig. 2d, top panel). However, we were surprised by the frequency of single nc11/12 nuclei displaying insulation at the border barcode. This frequency was smaller (10% with IS < 3.5), yet comparable to that observed in nc14 embryos, indicating that single nuclei displaying insulated TADs also exist in early developmental cycles before the emergence of ensemble TADs. Notably, TAD condensation does not seem to define the level of TAD insulation, as insulation scores and TAD volumes were uncorrelated both in nc11/12 and nc14 embryos (Fig. 2e, S2b).

To explore whether the presence of a TAD border was sufficient to split cells between UMAP populations, we color-coded each single nuclei in the UMAP by the insulation score at the border barcode (Figs. 2f and S2c). We observed a wide distribution of insulation scores for both UMAP populations and no strong correlation with any splitting in the UMAP. We observed a similar broad distribution of insulation scores for human HCT116 cells (Fig. S2c). Remarkably, nuclei with high insulation scores (i.e. low insulation) were common within the nc14 UMAP population (Fig. 2f). Similarly, single cells with high IS were common in untreated human HCT116 cells (Fig. S2c). Importantly, single nuclei in nc11/12 commonly displayed low IS values (i.e. high insulation), an observation that is mirrored by cohesin-depleted HCT116 single cells (Fig. S2c). In summary, only a minority of nuclei exhibit a strong border between TAD1 and doc-TAD in nc14 embryos, and a similarly sized population of insulated TADs is already present at earlier stages of development.

Finally, we applied Leiden clustering to shed further light into the distribution of single nuclei onto the UMAP space (see Methods). Following Occam's razor, we used a small number of Leiden clusters (6, Fig. 2g), however, use of more clusters lead to similar conclusions (Fig. S2d-e). Leiden clusters mapped approximately to nuclear cycle

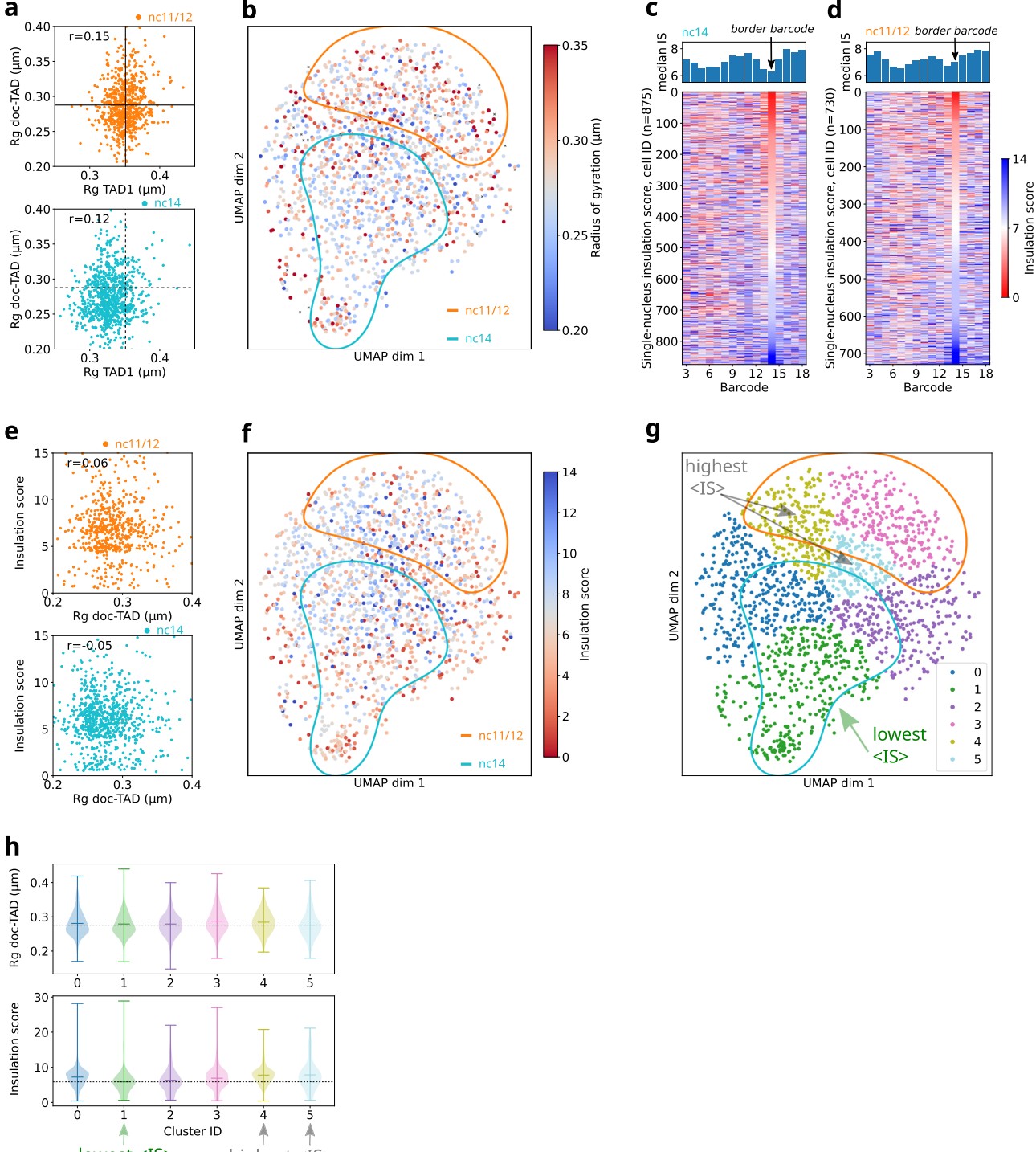

**Fig. 2 | Segregation of nuclei populations in the UMAP space is due to a combination of multiple architectural features. a** Top: scatter plot of the correlation between radii of gyration for doc-TAD and TAD1 in nc11/12 nuclei. Each point represents a single nucleus ($n = 713$). Horizontal and vertical lines indicate the mean of the distribution. $r$ is the Pearson correlation coefficient. Bottom: same but for nc14 ($n = 837$). Dashed lines indicate the mean of nc11/12. **b** Same UMAP as in Fig. 1i, color-coded by the radius of gyration of the doc-TAD. Blue and red correspond to small and large radii, respectively. **c** Top: Ensemble-average (median) profile of the insulation score for nc14. Black arrow indicates the border barcode. Bottom: single-nucleus insulation score. Each line corresponds to the insulation score profile of an individual nucleus, with values color-coded from red (small insulation score, i.e., strong insulation) to blue (large insulation score, weak insulation). The color scale is the same as in **e**. Nuclei are sorted according to their insulation score at the border barcode. Number of nuclei: $n = 875$. **d** Similar to **d**, but for nc11/12 nuclei. Number of

nuclei: $n = 730$. **e** Top: scatter plot of insulation score at the border barcode versus radius of gyration of the doc-TAD in nc11/12. Each point corresponds to a single nucleus ($n = 623$). $r$ is the Pearson correlation coefficient. Bottom: similar to the top, but for nc14 ($n = 770$). **f** Same UMAP as in Fig. 1i, color-coded by the insulation score at the border barcode. Blue and red correspond to small and large insulation scores, respectively. **g** Leiden cluster decomposition of the UMAP (resolution = 0.25). Leiden clusters are shown in different colors. Clusters with the lowest and highest IS mean values are highlighted by arrows. **h** Distributions of the radius of gyration and the insulation scores for each of the Leiden clusters shown in panel **g**. Markers of the violin plot indicate mean and extreme values of the distributions. Dashed horizontal lines highlight the level of the lowest mean. Number of nuclei for clusters 0–5: 354, 340, 280, 246, 238, and 147, respectively. Number of replicates for panels **a**–**h** = 2, number of embryos for nc11/12 = 7, for nc14 = 9. Source data for all panels are provided as a Source Data file.

clusters, however this mapping was not strict. The nc11/12 cluster was decomposed mainly into two Leiden clusters (3 and 4), while the nc14 cluster was decomposed mainly into three Leiden clusters (0, 1, 2). A final Leiden cluster (5) fell in between the nc14, nc11/12 clusters. To study the structural differences between Leiden clusters, we mapped the structural parameters used above (radius of gyration and insulation score, Fig. 2h). The radii of gyration were widely distributed and their means were similar amongst clusters, consistent with our previous analysis (Fig. 2a, b). The insulation scores were also widely distributed, but their means displayed more variability. In particular, the Leiden cluster with the lowest IS mean (Leiden cluster 1) overlapped widely with the nc14 cluster. This is consistent with the prevalence of low IS values in this region of the UMAP (Fig. 2f). In contrast, the Leiden clusters with the highest IS means were, to a large degree, located within the nc11/12 cluster (Leiden cluster 4) or in the region between nc14 and nc11/12 (Leiden cluster 5). This finding is consistent with these regions of the UMAP displaying slightly larger IS values (Fig. 2f). Finally, Leiden clusters exhibiting low and high IS mean values also contained a mix of nuclear cycles (Leiden clusters 2 and 5). All in all, this analysis shows that TAD insulation contributes to the distribution of single nuclei in the UMAP, however, this mapping is not one-to-one and other, multiple hidden structural parameters are likely necessary to describe the position of a single nuclei within the UMAP space.

### Transcriptionally active and inactive nuclei explore similar regions of the UMAP space

The doc genes are specifically expressed at nc14. Thus, we naturally wondered whether their expression contributed to the single-nucleus organization of chromatin at the *doc* locus. To address this question, we imaged DNA organization by Hi-M together with the detection of *doc1*-expressing nuclei by RNA-FISH in nc11/12 and nc14 embryos (Fig. 3a). The presence of *doc1* nuclear transcription hotspots was used to label each single nuclei in the embryo as *doc1* active or inactive (Fig. 3a). In a first attempt to correlate TAD organization and transcription, we calculated the median intra- and inter-TAD distances for individual active and inactive nuclei. As expected, intra-TAD distances were smaller than inter-TAD distances (Fig. 3b, Methods). Interestingly, both intra- and inter-TAD distances were higher for transcriptionally active nuclei. Similar trends were observed for the difference PWD map and for the distribution of doc-TAD volumes in active and inactive nuclei (Fig. S3a-b). Overall, these results indicate that the doc-TAD was −on average− slightly more decondensed and more segregated from TAD1 in actively transcribing nuclei than in inactive nuclei.

To determine whether this trend was also visible in single nuclei, we calculated the radius of gyration of the doc-TAD in each single nucleus and displayed it along the *doc1* activation pattern (Fig. 3c). We observed that the volume of the doc-TAD did not strongly correlate with the transcriptional status of single nuclei. Notably, nuclei with large and small TAD volumes were observed both in the active and inactive patterns (Fig. 3c, bottom panels). Thus, TAD volume seems to only poorly distinguish between single nuclei with different transcriptional states.

Different chromatin structures within TADs may give rise to similar TAD volumes, thus we turned to UMAP embedding to further explore the possible role of transcription in the 3D chromatin structure of single nuclei. For this, we embedded single nc11/12 and nc14 nuclei together, and color coded them according to their transcriptional status (Fig. 3d). *Doc1* is expressed at nc14, thus nc11/12 nuclei did not display RNA-FISH signals and, as a result, most active cells appeared within the region of the UMAP enriched in nc14 nuclei (Fig. 3d, cyan contour). Remarkably, active and inactive cells were homogeneously distributed across this region, consistent with transcription not playing a key role in determining the overall 3D conformation of single nuclei.

The genomic coverage of the oligopaint library used in these experiments was sufficient to visualize TADs in this locus, but not to detect specific regulatory interactions within the doc-TAD (Fig. 3d). Thus, we performed a similar analysis using a published dataset with a genomic coverage that enabled the detection of cis-regulatory interactions (Fig. S3c)[28]. Nc11/12 and nc14 cells occupied different regions of the UMAP embedding (Fig. 3e), consistent with the results obtained with the lower coverage oligopaint library (Fig. 1i). In addition, *doc1* active and inactive nuclei also intermingled within the nc14 pattern and did not segregate from each other. Consistent with these findings, TAD volumes were similar for active and inactive nuclei for the higher resolution library (Fig. S3d). All in all, these results suggest that the chromatin organization of active and inactive nuclei are indistinguishable at the ensemble and single nucleus levels.

### Single nucleus TAD insulation and intermingling are independent of transcriptional activity

Several lines of evidence suggest that TAD borders play a role in insulating enhancer-promoter interactions between neighboring TADs, however this role is currently under intense debate[14]. Thus, we wondered whether insulation at the single cell level affected the expression of *doc1*. For this, we plotted the single nucleus IS profiles ranked by IS at the border barcode together with transcriptional status (Fig. 4a). Notably, we did not observe a correlation between the transcriptional state of single nuclei and their insulation score (Fig. 4a). In fact, both active and inactive nuclei displayed low insulation scores (i.e. strong TAD border), and conversely in many active nuclei we failed to observe a clear single-nucleus boundary at the border barcode. These conclusions are supported by the ensemble IS profiles for active and inactive cells (Fig. 4b) which show that active cells are similarly insulated than inactive cells.

To test whether proximity of TAD1 enhancers to the *doc1* promoter influenced its transcription, we calculated the intermingling between TADs for active and inactive cells. TAD interminging was estimated by the demixing score, calculated by measuring the ratio between intra- to inter-TAD distances (see Methods). Notably, the distributions of demixing scores were very similar for both states of transcription (Fig. 4c). This result is inconsistent with physical proximity between TAD1 enhancers and the *doc1* promoter playing an important role in its transcriptional activation. As PWD distributions are broad, changes in short-range distances, i.e., a contact between loci, might be overlooked when only considering distances[56]. Therefore, we binarized the single nucleus PWD maps using a contact threshold of 0.25 $\mu$m (Fig. S4a) and tested whether the transcriptional state of single nuclei was linked to a change in the number of contacts between the two TADs (Fig. 4d). In fact, the distributions in the number of inter-TAD contacts were very similar for active and inactive nuclei, supporting the conclusions from our previous analysis using demixing scores.

All in all, these analyses show that at this locus TADs are insulated in a small proportion of nuclei, that insulation is not correlated with transcriptional activation at the single nucleus level, and that TAD intermingling is as common in active as in inactive nuclei. Thus, we conclude that transcription does not seem to play a key role in the structure of single nuclei as measured by TAD insulation and intermingling.

## Discussion

Population-average TADs require the presence of CTCF and cohesin in mammals[37,57–59], and can first be detected at the zygotic genome activation step in multiple species[22,23,60–62]. In this study, we used an imaging-based method that simultaneously provides developmental timing, transcriptional status, and snapshots of chromatin conformations in single nuclei during this developmental transition. This imaging method, combined with state-of-the-art single-cell analysis

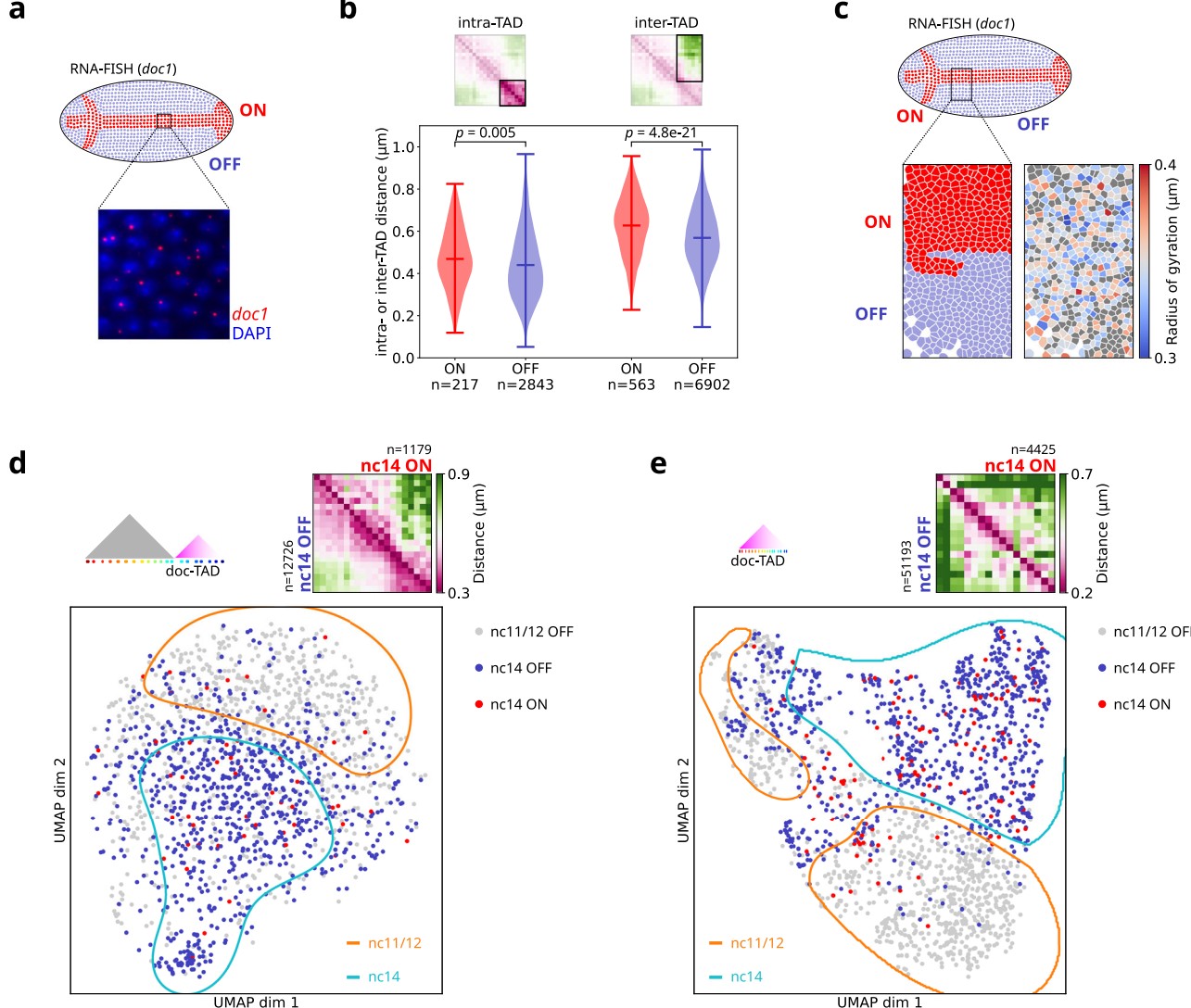

**Fig. 3 | Transcriptionally active and inactive nuclei explore a similar conformational space. a** Top: scheme of a *Drosophila* embryo in dorsal orientation. Nuclei transcribing *doc1* are indicated in red ("ON"), nuclei not transcribing *doc1* in blue ("OFF"). Bottom: DAPI-stained nuclei (blue) with *doc1* RNA-FISH spots indicating nascent transcripts (red). **b** Intra- and inter-TAD distance distributions shown as violin plots. For each nucleus, the median of the pairwise distances highlighted in the distance map on the top are calculated. Intra-TAD distances were calculated for the doc-TAD only, as shown on the region highlighted above. Distances for ON nuclei are shown in red, distances for OFF nuclei in blue. Markers indicate the mean and extreme values of the distribution. Number of nuclei (*n*) are indicated for all conditions. *p*-values calculated by a two-sided Welch's *t*-test. **c** Bottom left: Cell masks after DAPI image segmentation from a Hi-M experiment. The color indicates the transcriptional state. Bottom right: The same masks as on

the left are color-coded by the radius of gyration of the doc-TAD. Nuclei for which no radius of gyration could be calculated are in gray. **d** Top: Ensemble-average pairwise distance map for ON (top right half of the matrix) and OFF nuclei (bottom left half) for the low-resolution Hi-M library. Number of nuclei (*n*) as indicated in the figure. Bottom: Same UMAP as in Fig. 1i, color-coded by nuclear cycle and transcriptional state. nc11/12 OFF nuclei are in gray (*n* = 730), nc14 OFF (*n* = 823) in blue and nc14 ON in red (*n* = 52). **e** Similar to **b**, but for the high-resolution Hi-M library, that spans the doc-TAD with higher resolution. Number of nuclei: *n* = 926 (nc11/12 OFF), *n* = 801 (nc14 OFF), *n* = 133 (nc14 ON). For panels **b**, **d**, number of replicates = 2, number of embryos for nc11/12 = 7, for nc14 = 9. For panel **c**, **e**, number of replicates = 3, number of embryos for nc11/12 = 8, for nc14 = 29. Source data for panels **b**, **d**, and **e** are provided as a Source Data file.

techniques, provides insights into the roles that TAD insulation, TAD condensation and transcription play in shaping chromatin structure.

Chromatin organization at the TAD scale was previously shown to be highly variable both by FISH[34,35,37,38], and by polymer modeling simulations[5,39,40,63]. These models showed that chromatin forms globular TAD-like structures in the presence of attractive interactions between monomers in a TAD[39], or in the presence of loop extrusion by cohesin[64,65]. In contrast, chromatin behaves as a random-coil polymer in the absence of cohesin or in the absence of monomer-monomer interactions[39,64,65]. Notably, our results show that chromatin is similarly heterogeneous in the presence or absence of ensemble TADs, suggesting that intra-TAD interactions do not play a major role in

constraining the degree of variability of chromatin conformations. A reasonable explanation for this behavior, compatible with simulation and imaging data[34,35,39], is that intra-TAD interactions are transient and highly variable between cells, likely reflecting the binding dynamics of transcription factors and architectural proteins[66–69] and their cell-to-cell heterogeneity, as well as chromatin conformational dynamics.

Notably, despite this heterogeneity, the chromatin conformations of single nuclei before and after the emergence of ensemble TADs were distinct enough to occupy different regions in the 2D UMAP embedding of the conformational hyperspace. We observed similar results in datasets from untreated and cohesin-depleted human cells. Thus, tracing the path of chromatin in a single nucleus at the TAD-scale

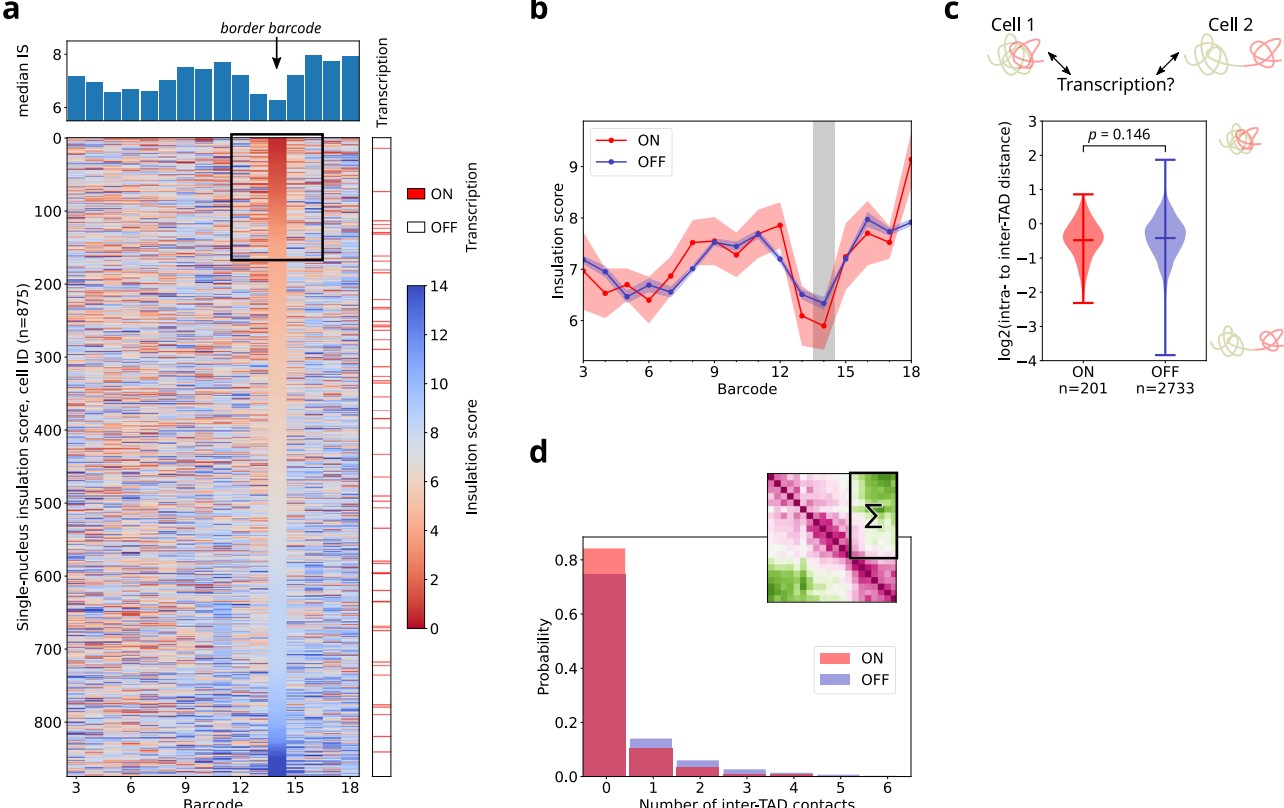

**Fig. 4 | Single-nucleus TAD insulation and intermingling with a neighboring TAD are independent of transcriptional activity. a** Top: Ensemble-average (median) profile of the insulation score for nc14 embryos. Bottom: Single-nucleus insulation score, number of nuclei: $n = 875$. Each line corresponds to the insulation score profile of an individual nucleus, with values color-coded from red (small insulation score, i.e., strong insulation) to blue (large insulation score, i.e., weak insulation). Nuclei are sorted according to their insulation score at the TAD border. The right-most lane indicates the transcriptional state of each nucleus, with active nuclei marked red. The black frame highlights the ~20% of nuclei with a high insulation (i.e. a low insulation score) at the ensemble TAD border. **b** Ensemble-average profile of the insulation score for active (red) and inactive (blue) nc14 nuclei shown in **a**. Circles indicate the median IS for each barcode. The red and blue

shaded bands represent the uncertainty as estimated by bootstrapping. The vertical gray bar indicates the border barcode. **c** Demixing score of TAD1 and doc-TAD for active (red) and inactive (blue) nuclei. Large values indicate a stronger intermingling of the two TADs as shown by the scheme to the right. Markers indicate the mean and extreme values of the distribution. Number of nuclei ($n$) as indicated. $p$-value calculated by a two-sided Welch's $t$-test. **d** Histogram of the number of inter-TAD contacts between TAD1 and doc-TAD. For each nucleus, the sum of contacts in the region highlighted in the distance map on the top is calculated. Active nuclei in red, inactive nuclei in blue. Number of nuclei: $n = 1179$ (ON), $n = 12,726$ (OFF). For panels **a**–**d**, number of replicates = 2, number of embryos for nc14 = 9. Source data for all panels are provided as a Source Data file.

and with relatively sparse sampling (20–50 barcodes) is sufficient to predict, with reasonable confidence, the presence of mechanisms leading to TAD formation. We insist, however, on the probabilistic nature of this prediction, given that (1) single nuclei conformations may fall, with low probability, into regions of the UMAP space displaying overlapping conformations; and (2) the regions of the UMAP space attributed to each population are not unequivocal.

These results suggested that structural parameters, such as TAD insulation or condensation, or transcriptional activity may be the main factors segregating single nuclei in the UMAP space. Surprisingly, this does not appear to be the case, as none of these factors played predominant roles in the segregation of single conformations in the UMAP space. For instance, while TADs tended to be more condensed in nc14 embryos, an important proportion of nc11/12 embryos also exhibited TADs condensed to similar levels. Surprisingly, only a fraction of nuclei displayed strong insulation between TAD1 and the doc-TAD in nc14 embryos, and a smaller but significant fraction of nuclei exhibited strong insulation in nc11/12 embryos. These results are consistent with Hi-C and imaging studies where TAD borders in *Drosophila* were shown to be variable between single nuclei[30,34]. In *Drosophila*, TAD boundaries are mainly occupied by architectural factors[4,13], and their appearance requires the binding of pioneering factors[22]. In this context, our results suggest that occupation of TAD borders: (1) is

stochastic, possibly due to the binding kinetics of architectural proteins and of components of the transcriptional machinery; (2) gradually increases during early development as these factors become more abundant. All in all, our results suggest that neither TAD condensation, nor TAD insulation can be used to predict whether a nucleus belongs to nc14 or nc11/12 embryos. We conclude that, instead of a single structural parameter, multiple combinatorial contributions from several structural properties may be required to assign single chromatin conformations to specific developmental stages with high confidence. This is consistent with a recent study showing that multiple, spatially distributed structural features are required to predict transcriptional activation in the BX-C domain during late *Drosophila* embryogenesis[70].

Cell-to-cell variations in the gene activation of genes within a TAD, either arising from controlled variations in transcriptional programs between cell-types or from stochasticity in transcription[71], could arguably contribute to the spatial distribution of single conformations within the UMAP space. Consistent with this idea and with previous evidence[34,41] active nuclei showed on average a slightly more decondensed chromatin architecture than inactive nuclei. However, despite these small differences, transcriptionally active and inactive nuclei explored overlapping conformational spaces and did not segregate from each other in the UMAP space. These results are consistent with recent studies showing that, during *Drosophila* development, the

population-averaged chromatin architectures of cells from different presumptive tissues display only overall small differences[27,28]. Thus, we conclude that transcription does not seem to play a dominant role in determining the chromatin structure of single nuclei at least during these early stages of *Drosophila* development.

The single nucleus snapshots of chromatin architecture occupied preferential regions in the UMAP space that depended on the developmental stage. How much of this conformational space can be explored by an individual nucleus during a biologically relevant timescale, i.e., one cell cycle? The succession of nuclear division cycles is fast during early development of *Drosophila*, with cell cycles lasting 10–12 min for nc11 and nc12 and at least 65 min for nc14[72]. Existing experimental approaches that dynamically track genomic loci[18,20,73] are currently not able to address the question of TAD-scale chromatin mobility directly as technological limitations in live-cell labeling so far have prevented dynamic visualization of more than two loci at the same time. Future studies that simultaneously track more than two genomic regions in vivo at high genomic (<10 kb) and optical (<150 nm) resolutions will be needed to determine whether chromatin within TADs dynamically explore their conformational space during interphase. Nevertheless, polymer physics predicts that a spatial neighborhood of 150–300 nm (roughly corresponding to a genomic region of 100 kb) can be extensively explored by a locus in 2–5 min[74,75]. This suggests that, despite short cell cycle times during early *Drosophila* development, a genomic region spanning a few TADs (as that investigated here) should be able to cover a substantial part of the conformational space outlined by single nucleus snapshots, leading to ergodicity at these short spatial scales.

Studies in mammalian cells have shown a non-zero probability for TAD-like domain boundaries at any locus[37] and a heterogeneity in the intermingling of two regions that are separated by a TAD border in the ensemble-average[42,43]. Our model system enabled the investigation of single nucleus variation in TAD insulation in intact embryos, where development progresses in a highly synchronous and orchestrated fashion in contrast to cell cultures. Thus, we expected the contribution of external sources of perturbation (e.g. differences in cell cycle stage or transcriptional activation within well-defined presumptive tissues) to be reduced. Nevertheless, we still observed that insulation between adjacent TADs is highly heterogeneous at the single nucleus level. In addition, we found that intermingling of the doc-TAD with the upstream TAD is independent of *doc1* activation. This lack of a strong insulation effect suggests that tissue- and time-specific enhancers carrying the right composition of transcription-activating factors, together with promoters bound by compatible sets of factors, are the main regulators of gene expression in this system. In this scenario, other nearby enhancers would only play a minor role, due to lack of compatible activating factors, even when they could be in spatial proximity to the promoter. Alternatively or in addition, a carefully balanced nonlinear relationship between enhancer-promoter contact frequency and transcriptional output could also explain the similar single nucleus insulation profile of active and inactive nuclei. Such non-linear models have been proposed by several groups recently[76–78]. These non-linear models have in common that gene activation is modeled as a multi-step process that makes repeated interactions between enhancer and promoter necessary before gene activation is achieved. Thus, EP contacts spanning a population-average TAD border could occur, but with a frequency that is not sufficient to effectively trigger transcription.

In the case of the *doc* genes, multiple enhancer elements (validated or putative) are distributed throughout the doc-TAD[28]. As the number of enhancers vastly outnumbers the number of genes[79], regulation of a gene by multiple enhancers seems to be the rule, not the exception. The classical gene regulation model assumes that stable, long-lasting EP contacts need to be formed, thus restricting the possibility for alternative chromatin configurations. By contrast, a highly flexible chromatin organization could be an advantage, as different enhancers could physically access the promoter region, thereby offering a way to ensure phenotypic robustness as first described for *Drosophila*[80,81] and later on in mammals[82]. This robustness would be especially important during early development when cells need to follow precise spatiotemporally gene-expression programs. It is interesting to note that a recent deep-learning analysis of chromatin tracing data suggest that chromatin structures linked to the transcriptional state of a gene are broadly distributed across the gene's regulatory domain and that individual enhancer-promoter interactions don't play a major role in defining the transcriptional activity of a gene[70].

Overall, our analysis of single-nucleus microscopy-based chromosome conformation capture data is compatible with a model of flexible chromatin organization within TADs that serves as a scaffold, with enhancer-bound transcription-activating factors encoding the logic that integrates multiple, potentially short-lived, interactions that can extend beyond domain borders defined from ensemble-averaged experiments.

## Methods

### *Drosophila* embryo collection

Embryos from fly stocks (Oregon-R w[1118]) were collected and fixed as previously described[28]. Briefly, following an O/N pre-laying period in cages with yeasted 0.4% acetic acid agar plates, flies were allowed to lay embryos during 1.5 h on the new plates at 25 °C. Embryos were then incubated for an extra 2.5 h to reach the desired developmental stage. Then embryos were dechorionated for 5 min with a solution of 2.6% bleach and thoroughly rinsed with water. Next, embryos were fixed for 25 min at the interface of a 1:1 mixture of fixation buffer (5 mL of 4% methanol-free formaldehyde in PBS and 5 mL of heptane). The bottom layer containing formaldehyde was then replaced by 5 mL methanol then vortexed for 30 s. Finally, embryos that sank to the bottom of the tube were rinsed three times in methanol then stored at −20 °C until further use.

### RNA-FISH probes preparation and hybridization

RNA probes for *doc1* containing a 5′ digoxigenin (DIG) modification were obtained by in vitro transcription as previously described in[28,49] (see sequences in Supplementary Table 2). In situ hybridization was performed as previously reported[28,49]. Briefly, embryos were dehydrated once with a solution of 1:1 methanol/ ethanol then five times in 100% ethanol for 5 min each on a rotating wheel then post-fixed with a solution of 5% formaldehyde in PBT (0.1% Tween-20 PBS) for 25 min. Embryos were incubated 4 times with PBT during 15 min followed by a permeabilization step in PBS 0.3% Triton for 1 h then rinsed three times 5 min with PBT. PBT was then replaced by 1:1 dilution of PBT with RHS (RHS = 50% formamide, 2X SSC, 0.1% Tween-20, 0.05 mg/ml heparin, 0.1 mg/ml salmon sperm). Next, embryos were incubated 10 min, 45 min and 1 h 15 min in three different RHS solutions at 55 °C under 900 rpm agitation. In parallel, 2 μL were added to 250 μL of RHS, denatured 2.5 min at 85 °C and placed on ice. The solution containing the probes was then added to embryos and incubated 16–20 h at 55 °C. The next day, the solution was removed and the embryos were washed four times at 55 °C for 30 min, one time with 1:1 PBT with RHS for 10 min and finally three times 20 min in PBT. Then, a saturation step was performed with 2X blocking solution (10X Blocking solution = 10% (w/V),100 mM Maleic acid, 150 mM NaCl, pH = 7.5) for 45 min, then incubated with PBT 1% $H_2O_2$ to eliminate the endogenous peroxidase activity and rinsed two times with PBT. Embryos were incubated O/N at 4 °C with a 1:500 dilution of sheep anti-DIG conjugated to POD (Sigma-Aldrich, catalog no. 11207733910) in PBT. The next day, embryos were washed 5 times with PBT 12 min each time. The tyramide signal amplification was performed with 5 μL of Alexa 488 coupled to tyramide dissolved in DMSO in 500 μL PBT for 30 min. Finally $H_2O_2$ was added to the tube for a final concentration of 0.012% during 30 min, then embryos were washed three times with PBT for 5 min.

## Hi-M libraries preparation and hybridization

Barcode positions and oligopaint-FISH libraries were previously described elsewhere[28] (Supplementary Tables 3-4 and Supplementary Data 1–2). In brief, the low-resolution library consisted of 20 barcodes and covered two adjacent TADs (3L:8889500..9045900, Release 6 reference genome assembly for *Drosophila melanogaster*), including the *doc* locus (Supplementary Table 3). The high-resolution library comprised 17 barcodes in the doc-TAD (3 L:8981462..9045820) (Supplementary Table 3). Each oligo in our library consists of five regions from 5′ to 3′: i- a 21-mer for the forward priming regions, ii- a single 32-mer (low-resolution) or two 20-mer unique readout regions (high-resolution library) separated by a spacer (AT), iii- a 35/41-mer genome homology region, iv- a single 32-mer (low-resolution) or one 20-mer unique readout region (high-resolution library) separated by a spacer (AT), v- a 21-mer for the reverse priming regions. The template oligonucleotide pools were ordered from CustomArray and then amplified as described in[36,49] (see Supplementary Data 1 for primer sequences). The amplification consists in: i- an emulsion PCR, ii- a large-scale PCR, iii- an in vitro T7 transcription followed by a reverse transcription, iv- an alkaline hydrolysis and a purification. The list of primary sequences are provided in Supplementary Data 2.

The hybridization of the amplified oligopaint FISH library was performed after the RNA-FISH protocol followed by TSA amplification as described in[28,36,49]. Briefly, embryos were treated with RNase for 2 h, permeabilized 1 h with 0.5% Triton in PBS with four different concentrations of Triton/pHM (pHM = 2X SSC, NaH 2PO 4 0.1 M pH = 7, 0.1% Tween-20, 50% formamide (v/v)): 20%pHM; 50%pHM; 80%pHM; 100%pHM for 20 min each on a rotating wheel. Then, 225 pmols of the pool were diluted in 30 μL of FHB (FHB = 50% Formamide, 10% dextran sulfate, 2X SSC, Salmon Sperm DNA 0.5 mg mL-1), the probes were added to the tube containing the embryos prior to heating it in a water bath at 80 °C during 15 min under a layer of mineral oil to avoid water evaporation. The tube was then incubated O/N at 37 °C. The next day, oil was removed and the embryos were washed two times during 20 min at 37 °C with 50% formamide, 2 × SSC, 0.3% CHAPS. Next, embryos were sequentially washed with four different concentrations of formamide/PBT: 40% formamide; 30% formamide; 20% formamide; 10% formamide for 20 min on a rotating wheel. Finally, embryos were washed with PBT, post-fixed with a solution of 4% formaldehyde in PBS and stored at 4 °C prior to imaging.

## Acquisition of Hi-M datasets

Experiments were performed on a homemade, wide-field epifluorescence microscope with previously described procedure[28]. Briefly, embryos were attached to a 10% poly-L-lysine- coated 40 mm coverslip and mounted into a FCS2 flow chamber (Bioptechs, US). The sample was flowed with 1800 μL of the fiducial readout probe (25 nM Rhodamine-labeled probe, 2 × SSC, 40% v/v formamide) for 15 min then washed with 2000 μL of washing buffer (2 × SSC, 40% v/v formamide), 1000 μL of 2× SSC then incubated 10 min with 0.5 mg/ml DAPI in PBS to stain nuclei. The imaging buffer (1× PBS, 5% w/v Glucose, 0.5 mg/ml glucose oxidase and 0.05 mg/ml catalase) was injected to prevent the bleaching of the fiducial readout probe. In a typical experiment, 10–15 embryos were immobilized and selected for imaging, using a field of view of 200 × 200 μm and 60 z-stacks (250 nm thick), and at the following excitation wavelengths: 405 (DAPI), 488 (RNA) and 561 nm (fiducial). Sequential injection was then performed with different secondary readout probes or adapters depending on the library used (low-resolution or high-resolution oligopaints library, respectively). For each round of hybridization, the sample was treated either with 50 nM of imaging oligonucleotide (Alexa Fluor 647 with a bisulfide bound) diluted in the washing buffer (low-resolution library) or with a mixture of 50 nM of the adapter and 50 nM of the imaging oligonucleotide diluted in washing buffer. Next, the sample was successively washed with 2000 μL of washing buffer, 1000 μL of 2× SSC

and incubated with the imaging buffer prior to the 3D image acquisition (same parameters as above) at 561 nm (fiducial) and 647 nm (barcode) laser illuminations. After imaging, the imaging oligonucleotide was extinguished using a chemical bleaching buffer (2 × SCC, 50 mM TCEP hydrochloride), and washed with 1000 μL of 2× SSC before a new hybridization cycle started.

## Image processing

The acquired dataset consisted of image stacks with 2048 × 2048 pixels and 60 slices (voxel size 0.106 × 0.106 × 0.250 μm³). Raw images supplied by the camera were in DCIMG format and were converted to TIFF using proprietary software from Hamamatsu. The TIFF images were then deconvolved using Huygens Professional v.20.04 (Scientific Volume Imaging, https://svi.nl). Further analysis was done using a homemade software pipeline written in python[28]. First, images were z-projected using sum (DAPI channel) or maximum intensity (barcodes, fiducials) projection. Then, for each hybridization round, the image of the fiducial channel was aligned to the reference fiducial image in a two-step process: (1) Global alignment by cross-correlation of the two images split in 8 × 8 non-overlapping blocks and averaging over the translation offset of all 64 blocks. (2) Local alignment in 3D of volumes each containing a single nucleus by cross-correlation.

Barcodes were segmented using a neural network (stardist)[83] specifically trained for the detection of 3D diffraction-limited spots produced by our microscope. To extract the position of the barcode with sub-pixel accuracy, a subsequent 3D Gaussian fit of the regions segmented by stardist was performed with Big-FISH (https://github.com/fish-quant/big-fish[84]). Barcode localizations with intensities lower than 1.5 times that of the background were filtered out.

Nuclei were segmented from projected DAPI images using stardist[83] with a neural network trained for detection of nuclei from *Drosophila* embryos under our imaging conditions. Barcodes were then attributed to single nuclei by using the XY coordinates of the barcodes and the DAPI masks of the nuclei. Finally, pairwise distance matrices were calculated for each single nucleus.

The transcriptional state of the nuclei was attributed by manually drawing polygons over the nuclei displaying a pattern of active transcription.

## Ensemble-average pairwise distance map

The ensemble pairwise distance map was calculated from the first maximum of the kernel density estimation for each pairwise distance distribution (Gaussian kernel, bandwidth 0.25 pixel, excluding pairwise distances larger than 4.0 μm), which is a robust approximation for the mode of the pairwise distance distribution. In case of the human HCT116 cell line data[37], the median of the pairwise distance distribution was used to calculate the ensemble pairwise distance map, following the approach of the original paper.

## Radius of gyration

The radius of gyration was calculated from the pairwise distances using $R_g^2 = \frac{1}{2N^2}\sum_{i,j}(d_{i,j})^2$, where $d_{i,j}$ is the pairwise distance between barcode $i$ and $j$. The number of pairwise distances ($N^2$) is adjusted in case a pairwise distance was not detected for a given cell. Pairwise distances above a threshold (1.0 μm) were set to NaN to remove outliers that would bias the radius of gyration towards large values. Cells with less than one third of all pairwise distances detected were excluded from analysis.

Calculation of the radius of gyration is based on the standard-resolution Hi-M library except for Fig. 3c, which uses the high-resolution Hi-M data as a slightly higher fraction of nuclei could be used to calculate the radius of gyration for this data.

## Pair correlation

The similarity of the chromatin organization between single nuclei was calculated by the pair correlation for all possible pairs of nuclei,

following[40]. In more detail, nuclei with a detection of less than 13 out of 20 barcodes (low-resolution library) or 11 out of 17 barcodes (high-resolution library) were excluded from analysis. The upper triangle of the sn pairwise distance maps was flattened to yield a vectorized representation. The Pearson correlation coefficient for all pairs of vectorized distance maps was calculated, using the reciprocal distance to be more sensitive to changes in small distances. Pairwise distances that were not detected in both corresponding nuclei were masked for the calculation of the correlation coefficient. A given pair of nuclei was skipped when the number of the overlapping detected pairwise distances is smaller than half of the full number of pairwise distances.

As a reference, the pair correlation of randomly shuffled pairwise distance maps was calculated. For this, for each pair one of the pairwise distance maps was reordered randomly ten times and the Pearson correlation coefficient was calculated as described above for each of the ten randomizations.

### UMAP embedding

The number of nuclei for different developmental time points (nc11/12 vs nc14) or different treatment (wt vs auxin treated) were roughly matched by adjusting the cutoff for excluding nuclei from analysis based on the number of detected barcodes. For the low-resolution library, nuclei were excluded when more than 7, or 6, out of 20 barcodes were not detected (nc11/12, or nc14, respectively). For the high-resolution library, nuclei were excluded when more than 5, or 2, out of 17 barcodes were not detected (nc11/12, or nc14, respectively). For the human HCT116 cell line data, nuclei were excluded when more than 5, or 1, out of 83 barcodes were not detected (auxin-treated, or wt, respectively).

For the remaining nuclei, single nucleus pairwise distances not detected were imputed with the corresponding value from the ensemble pairwise distance map. Then, the single nucleus pairwise distance maps were vectorized (see "pair correlation" above) and the different categories (developmental time points or treatments) were concatenated. Dimension reduction to two dimensions was achieved by unsupervised embedding, using the python implementation of "Uniform Manifold Approximation and Projection" (UMAP)[85]. Parameters were: $n\_neighbors = 50$, $min\_dist = 0.1$, $n\_epochs = 500$, $metric =$ "canberra".

### Density-based cluster boundaries in UMAP plots

To demarcate clusters in the UMAP plots, density-based boundaries were calculated. For this, the embedding was split according to the two categories (developmental time points or treatments) and the Gaussian kernel density estimate was calculated ($bandwidth = 0.5$). To exclude regions with no or only a few points, areas with a low density in each category were masked ($density\_threshold = 0.02$). Then, the difference between the two densities was calculated. Finally, the contour of areas with a difference in the density exceeding a threshold (0.02 for HCT116 and the low-resolution *Drosophila* data, 0.01 for the high-resolution *Drosophila* data) was used to demarcate clusters.

### Leiden clustering of UMAPs

The UMAP embeddings were used to perform a Leiden clustering using the scanpy toolkit[86]. For the computation of the k-nearest neighbors graph, the same number of neighbors as for the UMAP was used ($n\_neighbors = 50$). Additional parameters were: $n\_pcs = 0$, metric = "euclidean". For the resolution parameter of the Leiden clustering: see legends of Figs. 2g, and S2d–e.

### Insulation score

If indicated, the same selection of nuclei and imputation of missing sn pairwise distances as for the UMAP embedding was performed. Otherwise the raw data was used. To calculate the sn insulation score, we followed an approach similar to the one used for bulk Hi-C contact maps[55]. In short, a square of 2-by-2 barcodes is moved parallel to the diagonal of the pairwise distance map and the inverse of the pairwise distances in this square are summed. This yields a profile of the insulation score per nucleus with lower insulation score values corresponding to a higher insulation.

### Intra- and inter-TAD distances and demixing score

To get average intra- and inter-TAD distances for each nucleus, the median of all pairwise distances in the doc-TAD or between TAD1 and doc-TAD were calculated. Distances above a threshold (1.0 μm) were excluded and medians were calculated only when at least 3 intra- or inter-TAD distances were detected for a nucleus. The demixing score is calculated per nucleus as the log2 ratio of the median intra- and inter-TAD distances.

### Inter-TAD contacts and 3-way interactions

Two barcodes were considered in contact when their distance was <250 nm. The number of inter-TAD contacts per nucleus was obtained by counting the number of contacts in the inter-TAD region in the contact map.

### Statistics

The *p*-values in Figs. 3b, 4c, and S1e were calculated by a two-sided Welch's *t*-test.

### Reporting summary

Further information on research design is available in the Nature Research Reporting Summary linked to this article.

## Data availability

The data that support this study are available from the corresponding author upon reasonable request. Single nucleus pairwise distance matrices generated in this study have been deposited in the Zenodo database under https://doi.org/10.5281/zenodo.6861355 [https://zenodo.org/record/6861355]. Previously published datasets used in this study include GSE86966, GSE65441, and GSE103625), and are listed in Supplementary Table 1. Source data are provided with this paper and are also accessible at https://doi.org/10.5281/zenodo.6861355 [https://zenodo.org/record/6861355].

## Code availability

The code used in this manuscript is accessible at: https://github.com/NollmannLab/Goetz_etal. Data were collected using our home-made Hi-M software package available at https://github.com/NollmannLab/HiMacquisitionSoft. For permanent link, see https://doi.org/10.5281/zenodo.6811576.

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

## Acknowledgements

This project was funded by the European Union's Horizon 2020 Research and Innovation Program (Grant ID 724429) (M.N.). We acknowledge the Bettencourt-Schueller Foundation for their prize 'Coup d'élan pour la recherche Française', the France-BioImaging infrastructure supported by the French National Research Agency (grant ID ANR-10-INBS-04, "Investments for the Future"), and the Drosophila facility (BioCampus Montpellier, CNRS, INSERM, Univ Montpellier, Montpellier, France). M.G. was funded by the Deutsche Forschungsgemeinschaft (DFG, German Research Foundation) - project ID 431471305. O.M. is supported by an FRM PhD fellowship.

## Author contributions

M.G. and M.N. conceived the study and the design. S.M.E. and O.M. acquired the data. M.G., O.M., S.M.E., and M.N. analyzed the data. M.G., M.N. and J.-B.F. provided the software. M.G., O.M., and M.N. interpreted the data. M.N. and M.G. wrote the manuscript. M.G. and M.N. did the visualization of the study. M.N. supervised the study and acquired funds.

## Competing interests

The authors declare no competing interests.
