## [Peer Review File · Nature Communications]

REVIEWER COMMENTS

Reviewer #1 (Remarks to the Author):

Gotz and colleagues report heterogeneity in chromatin conformation during early drosophila embryogenesis. The authors applied a DNA-FISH based method to capturing chromatin organization at single cell level. Although this approach not leading to visualization of chromatin structure in detail, the authors suggest indistinguishable active/inactive transcription status with chromatin conformation at the TAD scale from analyses using multiple parameters. The manuscript clearly addresses that transcription is not a major factor to determine the chromatin structure. This study is noteworthy and informative to research chromatin compartments related to transcriptional status.

Major concerns)

(1) The authors used the Hi-M method to figure out chromatin conformation. Recently, this kinds of method based on microscope and application of FISH have emerged to capture chromatin structure. Authors should address the pros/cons of Hi-M compared to previous method briefly or emphasize the reason why Hi-M method is useful to this study.

(2) In Fig 1g, the authors used HCT116 to support that TAD borders status of nc11/12-nc14 is similar with cohesion depleted status. But, there is a concern that cohesion complex would be active working during cell division, especially embryogenesis. The authors should address the reason why they choose HCT116 to compare their results and describe more in detail.

(3) In this study, the authors used several parameters to state differences between division cell cycle. To improve readability, authors should represent summarized table or schematic diagram by division cycle or time line.

Minor concerns)

(1) In second paragraph of results, authors should specify the functions of doc 1 gene to understand the reason why they use doc1 expression in overall study.

Reviewer #2 (Remarks to the Author):

In this manuscript Götz et al. seek the sources of the observed discrepancy between bulk and single-nuclei genome conformation analyses. While bulk analyses, such as Hi-C, revealed that the genome is organized in invariant and discrete topologically associating domains (TADs), single-cell analyses such as single-cell Hi-C or imaging-based approaches, demonstrate a great deal of variability in chromatin organization between individual nuclei. What is the importance of TADs and the observed cell-to-cell heterogeneity to transcriptional regulation is an opened and heated question in the field.

Here, Götz et al. address these problems by implementing Hi-M, an imaging-based chromosome conformation method they have previously developed, to study single-nuclei chromatin organization in early *Drosophila* embryos. During *Drosophila* embryogenesis TAD boundaries first emerge around nuclear cycle 14, at the onset of zygotic transcription. The authors exploit this transition to compare the 3D chromatin organization of a genomic region containing two TADs, TAD1 and doc-TAD, in individual nuclei before (cycle 11/12) and after (cycle 14) the emergence of TADs.

They first show that individual nuclei, at both cycle 11/12 and cycle 14, display a great deal of chromatin conformation heterogeneity. Despite this heterogeneity, as a population, the nuclei show clear TAD emergence at cycle 14. They next test if nuclei at cycle 11/12 and at cycle 14 explore the same conformational space by using a clever unsupervised UMAP-ing method. They demonstrate that despite the large heterogeneity detected in both stages, the nuclei are segregated into two distinct clusters according to their nuclear cycle, suggesting that chromatin conformation in single nuclei changes considerably during early *Drosophila* embryogenesis.

To test the factors that influence these phenomena they analyzed: (1) TAD volume by measuring the radius of gyration of the two TADs; (2) TAD insulation by calculating insulation scores; and (3) transcriptional states by combined DNA and RNA-FISH. However, none of these factors could explain, on their own, the observed heterogeneity in chromatin organization and the segregation of single nuclei on the conformational maps according to their developmental stage. Finally, they tested the relationship between TAD insulation and active transcription and found no correlation between them. They conclude that transcription does not play a key role in the chromatin structure of single nuclei.

I think that despite the negative results, this study is very important and will advance the field towards a solution of this complicated and multifactorial problem. The experiments and analyses have been carried out carefully and to an exceptionally high standard. The results, which are presented clearly in

text and figures, support the basic findings. I therefore recommend this manuscript for publication in Nature Communication. I have only a few minor comments:

1. In Supp. Figure 1c-d, please indicate the axes titles more clearly.
2. In Figure 1d-e, comparing the integral of the curves at the positive vs negative sides of the plots will strengthen the statement that the distribution of paired correlations was skewed toward the positive values.
3. While the goal and the conclusion of the analysis presented in Figure 2a are clear, the reader will benefit a better explanation of the analysis itself and how the authors drew their conclusion out of it.
4. In Figures 3d-e, I wonder why nc11/12 are marked in gray and not according to their transcriptional state. I understand that the doc genes are not expressed in this stage, but it is not clear if the combined DNA-FISH and RNA-FISH was done in both stages. If it was done only on stage 14 embryos, I think it will be less confusing to display only nc14 data or at least state so in the text.
5. Please cite Frankel et al. 2010 and Perry et al. 2010 when discussing enhancer redundancy and robustness. These studies made this discovery in *Drosophila* almost a decade before it was demonstrated in mammals.

Reviewer #3 (Remarks to the Author):

The Nollmann lab had previously published a DNA FISH method, Hi-M, which can generate 3D contact maps for barcoded sites distributed along a TAD-scaled locus in the chromatin of whole-mounted *Drosophila* embryos. The 3D contact maps allow one to infer the 3D structure of the chromatin organization on both the single-cell and the ensemble scale.

In this paper, Gotz et al. applied Hi-M to a different chromatin locus that spans two TAD regions, TAD1 and doc-TAD. Data from two different developmental stages of the *Drosophila* embryo, nc 11/12 and nc 14, were collected and analyzed. The authors recovered ensemble structural heterogeneity in chromatin organization in the TAD1 / doc-TAD locus both within the same and between different developmental stages, and were able to visualize this heterogeneity via a UMAP embedding, which shows two clusters for the nc 11/12 and nc 14 data. The authors then measured a few extra parameters from the Hi-M data, including the TAD radius of gyration and the insulation score at the locus between the two TAD regions, and found that neither of these parameters alone seem to explain the UMAP embedding. Furthermore, the authors measured expression on-off of the doc1 gene via RNA FISH, and found that both intra- and inter-TAD distances were higher in transcriptionally active nuclei. However, transcriptional active nuclei were distributed randomly in the UMAP space. The authors further showed

that the transcriptional activity seemed to not affect the TAD radius of gyration, TAD intermingling, insulation score at the TAD border, or inter-TAD contacts.

We think this paper is suitable for publication in Nature Communications. However, there are some major concerns that the authors should address before this paper is accepted. Our concerns can be grouped into three main categories: Clarity, Execution, and Minor Concerns.

Execution

Our major concern with the execution of this paper is related to how the authors utilized and interpreted the UMAPs of contact matrices.

---- In a major portion of the paper and the figures, the authors tried to explain the UMAP clustering with a single structural parameter, such as the radius of gyration or TAD insulation. However, it is unclear if it is reasonable to assume that a single chromatin structural parameter can be expected to explain the UMAP of contact matrices. As an analogy, consider the analysis of single-cell RNA sequencing data, in which it is common to use UMAP to visualize the differences between sets of cells. In this analogy the authors would be asking if a single parameter, such as the total transcript counts per cell, could explain the differences between clusters of cells. Certainly there may be instances where this is the case; however, in most cases this is not possible. Could the authors justify their approach or provide some alternative approaches to interpret the UMAPs?

---- We also have a question related to the division of UMAP clusters. In the UMAPs displayed in this paper, closed circular boundaries were drawn around each cluster (nc 11/12 or nc 14). However, in many of the UMAPs, it is hard to see how these boundaries were drawn since the UMAP clusters (for example, nc 11/12 vs. nc 14) were not well separated. Without these boundaries, the underlying distribution seems more intermixed to the eye. Can the authors specify how these cluster boundaries were calculated or defined?

---- Related to the previous point: In the discussion associated with Fig 1h: The authors claimed that most of the cells from nc 11/12 and nc 14 segregated into their own UMAP clusters. However, there is certainly some intermixing in the UMAP space: many of the nc 11/12 nuclei, in particular, are present within the boundaries drawn for nc 14. Can the authors provide the exact percentages of how many cells crossed over to the other cluster, like they did for the discussion associated with Fig 1g?

---- In the discussion associated with Fig 2a: The authors claimed that averaged distance matrices of the nuclei from the east and west parts of the UMAP (top two in that column) were similar to Fig 1b.

However, they looked really different from either the nc 11 or the nc 14 matrix in Fig 1b. and this similarity is really not visible to the eye. Can the authors provide a correlation number between the distance matrices to demonstrate this similarity? Similarly, it would be great if the authors can provide a correlation number to show that the south and north portions from the UMAP are similar to Fig 1c, as they also claimed.

--- And, related to the previous point: We are also uncertain whether dividing up the UMAP into North, South, East, and West is truly meaningful, since the axes on the UMAP have no real biological meaning, and can be easily rotated or flipped with some minor changes in the input data or embedding parameters. Related to the general theme of this section, can the authors come up with some alternative ways of interpreting their UMAPs? For example, can the authors explore Leiden / Louvain clustering on the UMAPs and see if they can retrieve distinct nuclei clusters? How Do the Leiden / Louvain clusters overlap with the nc 11/12 vs. nc 14 groups, and how do the structural parameters (radius of gyration, etc.) map onto the Leiden / Louvain clusters and / or the cell cycle groups?

--- This is a stand-alone point not related to UMAPs: In the discussion associated with Fig 4d.: the authors picked 0.25 um as the contact threshold, and discovered that the distribution of the number of contacts are similar with or without doc1 activation. Can the authors explain how they selected this threshold, or whether they had experimented with other contact distance thresholds to reach the same conclusion? Another concern here is whether the selection of barcodes is too sparse to capture enhancer-promoter interactions between the two TAD regions

Clarity

For clarity, we have some comments about the interpretation of the figures and disambiguation of the text.

--- In the discussion associated with Fig 1c: The authors claimed that the most notable difference between the two different nuclear cycles is that the distance between TAD1 and doc-TAD barcodes increased in nc14. However, it is difficult to see this overall trend within the pink box in Fig 1c, as there seem to be as many dark pink squares (increases) as there are dark blue squares (decreases) in the pink box. Can the authors clarify what they mean, or find alternative ways to visualize these data to support their claim?

---- Under the section that begins with "Single nuclei displaying insulated TADs are common ...", last paragraph, last sentence, the authors said that "this population of insulated TAD is already present at earlier stages of development ..." Were the authors trying to make the point that the nuclei that are insulated in nc14 are the same ones that are already insulated in nc 11/12 and then carried over to nc 14, or simply the percentages are similar in both nuclear cycles? The authors might want to modify the statement to disambiguate

---- We find the last paragraph in the Discussions sections, the conclusion of this paper, too speculative and not fully supported by the data shown. In particular, we are confused by the statement "flexible chromatin organization ... that can extend beyond the domain borders", since it seems to contradict the discussion associated with Fig 4: that the interaction between TAD1 and doc-TAD is not affected by doc1 activation, and therefore showing no evidence of enhancer-promoter interactions between neighboring TADs. Can the authors clarify what they mean or disambiguate?

Minor Concerns

---- This paper focused on the structure of only one chromatin locus. However, the title, "Multiple parameters shape the 3D chromatin structure of single nuclei", suggests a general conclusion. We are concerned that the discoveries from a single chromatin locus might not be representative of the entire nucleus, and suggest that the authors modify the title

---- Fig 1b, 1c: The font on the color bar is too small and unlegible, making it hard to tell which color means higher distance

---- Fig 3b: The distance maps lack a color bar. In addition, the authors should specify that by "intra-TAD", they mean doc-TAD not TAD1

---- Fig S1f: The authors should specify which color in the UMAP means cohesin-treated, and which one is untreated

---- We found the font size to be too small for some figure legends and axes

Reviewer #1 (Remarks to the Author):

Gotz and colleagues report heterogeneity in chromatin conformation during early drosophila embryogenesis. The authors applied a DNA-FISH based method to capturing chromatin organization at single cell level. Although this approach not leading to visualization of chromatin structure in detail, the authors suggest indistinguishable active/inactive transcription status with chromatin conformation at the TAD scale from analyses using multiple parameters. The manuscript clearly addresses that transcription is not a major factor to determine the chromatin structure. This study is noteworthy and informative to research chromatin compartments related to transcriptional status.

Major concerns)

1.1 *The authors used the Hi-M method to figure out chromatin conformation. Recently, this kind of method based on microscope and application of FISH have emerged to capture chromatin structure. Authors should address the pros/cons of Hi-M compared to previous methods briefly or emphasize the reason why Hi-M method is useful to this study.*

We thank the reviewer for raising this important point. We now provide a description of the limitations of other methods, and the advantages of HiM with respect to them to emphasize the reason why HiM was useful in this study. The introduction now reads (revised text in blue):

“(…)How these structural properties relate to single-cell chromatin structures and to transcriptional regulation is currently unclear. *This is in part because of limitations in ensemble sequencing-based chromosome conformation capture (3C) methods that cannot simultaneously capture chromosome structure and transcriptional status in single cells, and to limitations in conventional fluorescence in situ hybridization (FISH) techniques that can only visualize a very limited number of genomic loci at once.* Here, we investigated chromatin organization in single-nuclei before and after emergence of TADs during *Drosophila* embryogenesis. For this, we resorted to Hi-M, a microscopy-based chromosome conformation capture *method that simultaneously detects the 3D position of multiple genomic loci and transcriptional status in single cells.*”

1.2 *In Fig 1g, the authors used HCT116 to support that TAD borders status of nc11/12-nc14 is similar with cohesion depleted status. But, there is a concern that the cohesin complex would be actively working during cell division, especially embryogenesis. The authors should address the reason why they choose HCT116 to compare their results and describe more in detail.*

We apologize for the lack of clarity in our original submission. In our manuscript, we use HCT116 as a validation of the ability of our UMAP-based approach to visualize how single-cell chromatin conformations change between two different conditions. In HCT116 cells (wild-type and cohesin depleted), we observe the splitting of chromatin structures into two clusters in the UMAP. These clusters correspond to wild-type and cohesin-depleted cells.

In *Drosophila*, we observe a comparable splitting of the conformations of single-nuclei into two clusters that are less well defined than for HCT116. These two clusters correspond to nc11/12 and to nc14 nuclei. It is known that during *Drosophila* embryogenesis TADs emerge

at nc14 and are not present in earlier nuclear cycles ¹. However, the mechanisms by which TADs emerge in *Drosophila* are not thought to involve cohesin/CTCF, as in mammals ^{2,3}. Therefore, we do not imply that *Drosophila* cohesin is involved in the segregation of nc11/12 from nc14 nuclei that we observe.

We amended the text as follows:

“(…) Therefore, we conclude that despite a large degree of heterogeneity, the chromatin structures of single nuclei change considerably during the early cycles of *Drosophila* embryogenesis and occupy distinct conformational spaces. The mechanisms driving the segregation of single chromatin conformations in HCT116 cells and in *Drosophila* nuclei are likely to be different, as CTCF/cohesin does not seem to be the main actor in the formation of TADs in *Drosophila* ³.”

1.3 *In this study, the authors used several parameters to state differences between division cell cycles. To improve readability, authors should represent a summarized table or schematic diagram by division cycle or time line.*

In this study we investigate the differences in chromatin organization between two different nuclear cycles (nc11/12) and nc14 (Figures 1-2), and between transcriptionally ON and OFF single nuclei at nc14 (Figures 3-4). The comparison of structures of single nuclei is complex as each single nuclei structure contains hundreds of parameters. This comparison required the novel adaptation of embedding tools (UMAPs). We do not see how we could summarize these results in a table, but if the reviewer has specific suggestions we would be more than happy to follow them.

We also envisaged using a schematic diagram to display differences in chromatin structures between nuclei. These structures are highly variable from cell to cell, and the ‘ensemble structure’ does not actually have much meaning in this context. We fear that a schematic representation could be misleading in that it would convey the message that structures are static and different between nuclear cycles, whilst our analysis shows that these structures are highly variable. Therefore, we would prefer not to use a diagram to summarize these results.

Minor concerns)

1.4 *In the second paragraph of results, authors should specify the functions of doc 1 gene to understand the reason why they use doc1 expression in overall study.*

To address this issue, we expanded our explanation in the second paragraph of the results, as follows:

“The doc-TAD contains three developmental genes (*doc1*, *doc2*, and *doc3*) ⁴ that are specifically activated at nc14 in the dorsal ectoderm ⁵ (Fig. 1b). The doc genes encode functionally redundant T-box transcription factors essential for the development of the amnioserosa and cardiogenesis ⁶, and display similar, well-defined, cross-shaped expression patterns in the blastoderm embryo (nc11 to nc14).”

Reviewer #2 (Remarks to the Author):

In this manuscript Götz et al. seek the sources of the observed discrepancy between bulk and single-nuclei genome conformation analysis. While bulk analyses, such as Hi-C, revealed that the genome is organized in invariant and discrete topologically associating domains (TADs), single-cell analyses such as single-cell Hi-C or imaging-based approaches, demonstrate a great deal of variability in chromatin organization between individual nuclei. What is the importance of TADs and the observed cell-to-cell heterogeneity to transcriptional regulation is an open and heated question in the field.

*Here, Götz et al. address these problems by implementing Hi-M, an imaging-based chromosome conformation method they have previously developed, to study single-nuclei chromatin organization in early *Drosophila* embryos. During *Drosophila* embryogenesis TAD boundaries first emerge around nuclear cycle 14, at the onset of zygotic transcription. The authors exploit this transition to compare the 3D chromatin organization of a genomic region containing two TADs, TAD1 and doc-TAD, in individual nuclei before (cycle 11/12) and after (cycle 14) the emergence of TADs.*

*They first show that individual nuclei, at both cycle 11/12 and cycle 14, display a great deal of chromatin conformation heterogeneity. Despite this heterogeneity, as a population, the nuclei show clear TAD emergence at cycle 14. They next test if nuclei at cycle 11/12 and at cycle 14 explore the same conformational space by using a clever unsupervised UMAP-ing method. They demonstrate that despite the large heterogeneity detected in both stages, the nuclei are segregated into two distinct clusters according to their nuclear cycle, suggesting that chromatin conformation in single nuclei changes considerably during early *Drosophila* embryogenesis.*

To test the factors that influence these phenomena they analyzed: (1) TAD volume by measuring the radius of gyration of the two TADs; (2) TAD insulation by calculating insulation scores; and (3) transcriptional states by combined DNA and RNA-FISH. However, none of these factors could explain, on their own, the observed heterogeneity in chromatin organization and the segregation of single nuclei on the conformational maps according to their developmental stage. Finally, they tested the relationship between TAD insulation and active transcription and found no correlation between them. They conclude that transcription does not play a key role in the chromatin structure of single nuclei.

*I think that despite the negative results, this study is very important and will advance the field towards a solution of this complicated and multifactorial problem. The experiments and analyses have been carried out carefully and to an exceptionally high standard. The results, which are presented clearly in text and figures, support the basic findings. I therefore recommend this manuscript for publication in *Nature Communication*. I have only a few minor comments:*

We thank the referee for their thorough review of the manuscript and for their comments to improve it. We have addressed these points by implementing changes to the manuscript and figures that are described below.

2.1. *In Supp. Figure 1c-d, please indicate the axes titles more clearly.*

To address this issue, we added x and y-axis labels to both panels ('barcode #'), barcode numbers and symbols as in Fig. 1c, and panel titles ('PWD histograms'). The revised figure panels are as follows:

2.2. *In Figure 1d-e, comparing the integral of the curves at the positive vs negative sides of the plots will strengthen the statement that the distribution of paired correlations was skewed toward the positive values.*

We agree with the reviewer and therefore calculated these integrals, which are as follows:

- nc11/12: fraction negative = 0.41; fraction positive = 0.59
- nc14: fraction negative = 0.41; fraction positive = 0.59

The revised text now reads:

“Nevertheless, the distribution was skewed towards positive values (Figs. 1d-e, black line indicates median of the distribution), and the integral of the curve was lower for negative than for positive pair-correlations (0.6 versus 0.4 for both Figs. 1d-e), indicating that only a small proportion of nuclei displayed similar chromatin conformations.”

2.3. *While the goal and the conclusion of the analysis presented in Figure 2a are clear, the reader will benefit from a better explanation of the analysis itself and how the authors drew their conclusion out of it.*

Referee 3 argued in comment 3.6 that this analysis may not be truly informative as the directions of the UMAP do not have any biological meaning. For this reason, we removed this analysis from the revised manuscript. See answers to 3.5-3.6 below.

2.4 *In Figures 3d-e, I wonder why nc11/12 are marked in gray and not according to their transcriptional state. I understand that the doc genes are not expressed in this stage, but it is not clear if the combined DNA-FISH and RNA-FISH was done in both stages. If it was done only on stage 14 embryos, I think it will be less confusing to display only nc14 data or at least state so in the text.*

We apologize for the confusion. We have performed RNA-FISH in both stages, but could not detect cells displaying transcription spots in nc11/12 nuclei. We could have labeled these nuclei in blue (as for nc14 OFF nuclei) but at the risk of confusing nc11/12 and nc14 OFF cells. To address the comment of the referee, we have kept the same color scheme, but modified the figure legends (now stating 'nc11/12 OFF') and the text to avoid any misunderstanding. The revised paragraphs read as follows:

"Naturally, we wondered whether transcription contributed to the single-nucleus organization of chromatin at the *doc* locus. To address this question, we imaged DNA organization by Hi-M together with the detection of *doc1*-expressing nuclei by RNA-FISH in nc11/12 and nc14 embryos (Fig. 3a). (...)"

"(...) *Doc1* is expressed at nc14, thus nc11/12 nuclei did not display RNA-FISH signals and, as a result, most active cells appeared within the region of the UMAP enriched in nc14 nuclei (Fig. 3d, cyan shape). (...)"

2.5. *Please cite Frankel et al. 2010 and Perry et al. 2010 when discussing enhancer redundancy and robustness. These studies made this discovery in Drosophila almost a decade before it was demonstrated in mammals.*

We thank the reviewer for pointing this out. We have now added these citations to the Discussion section. The revised sentence now reads:

"By contrast, a highly flexible chromatin organization could be an advantage, as different enhancers could physically access the promoter region, thereby offering a way to ensure phenotypic robustness as first described for *Drosophila*^{7,8} and later on in mammals⁹."

Reviewer #3 (Remarks to the Author):

The Nollmann lab had previously published a DNA FISH method, Hi-M, which can generate 3D contact maps for barcoded sites distributed along a TAD-scaled locus in the chromatin of whole-mounted Drosophila embryos. The 3D contact maps allow one to infer the 3D structure of the chromatin organization on both the single-cell and the ensemble scale.

In this paper, Gotz et al. applied Hi-M to a different chromatin locus that spans two TAD regions, TAD1 and doc-TAD. Data from two different developmental stages of the Drosophila embryo, nc 11/12 and nc 14, were collected and analyzed. The authors recovered ensemble structural heterogeneity in chromatin organization in the TAD1 / doc-TAD locus both within the same and between different developmental stages, and were able to visualize this heterogeneity via a UMAP embedding, which shows two clusters for the nc 11/12 and nc 14 data. The authors then measured a few extra parameters from the Hi-M data, including the TAD radius of gyration and the insulation score at the locus between the two TAD regions, and found that neither of these parameters alone seem to explain the UMAP embedding. Furthermore, the authors measured expression on-off of the doc1 gene via RNA FISH, and found that both intra- and inter-TAD distances were higher in transcriptionally active nuclei. However, transcriptional active nuclei were distributed randomly in the UMAP space. The authors further showed that the transcriptional activity seemed to not affect the TAD radius of gyration, TAD intermingling, insulation score at the TAD border, or inter-TAD contacts.

We think this paper is suitable for publication in Nature Communications. However, there are some major concerns that the authors should address before this paper is accepted. Our concerns can be grouped into three main categories: Clarity, Execution, and Minor Concerns.

We thank the reviewer for the excellent summary of the results we present and for the constructive criticism to improve the manuscript. We provide a point-by-point response to these comments below.

Execution

3.1 *Our major concern with the execution of this paper is related to how the authors utilized and interpreted the UMAPs of contact matrices.*

We hope our answers below reassure the reviewer on how we interpreted the UMAP analysis of pairwise distance maps from single-nuclei.

3.2 *In a major portion of the paper and the figures, the authors tried to explain the UMAP clustering with a single structural parameter, such as the radius of gyration or TAD insulation. However, it is unclear if it is reasonable to assume that a single chromatin structural parameter can be expected to explain the UMAP of contact matrices. As an analogy, consider the analysis of single-cell RNA sequencing data, in which it is common to use UMAP to visualize the differences between sets of cells. In this analogy the authors would be asking if a single parameter, such as the total transcript counts per cell, could explain the differences between clusters of cells. Certainly there may be instances where this is the*

case; however, in most cases this is not possible. Could the authors justify their approach or provide some alternative approaches to interpret the UMAPs?

We apologize for not providing a clearer explanation of the rationale behind our approach. We understand the concern of the referee with our approach that can seem, at first glance, simplistic. The main reasoning behind our approach was as follows:

- A single parameter can indeed explain, to a considerable extent, the distribution of single nuclei in the UMAP, as shown for the developmental timing (Fig. 1).
- Chromatin condensation is a rough but direct measure of chromatin structure, and was thus often used to characterize how chromatin conformation changes with epigenetic state (e.g. ¹⁰), transcriptional activation (e.g. ^{11–15}), cell cycle (e.g. ^{16–19}), etc. Thus, it was reasonable to study whether the distribution of single nuclei on UMAPs is indicative of different chromatin condensation states. Our main finding is that TADs get globally condensed from nc11/12 to nc14, however, there are large variations in condensation level within nuclear cycles. Thus, TAD condensation plays a role in the global separation of nuclei within nuclear cycle clusters but does not seem to contribute considerably to the positioning of nuclei in the UMAP.
- In *Drosophila*, TADs arise in nc14 and are not observed in previous nuclear cycles ^{1,14,20}. The insulation score is typically used as a tool to monitor the emergence of TADs. Thus, it was reasonable to hypothesize that the distribution of nc14 and nc11/12 nuclei on the UMAP could be influenced by the insulation score (IS) of the TAD boundary in the doc locus. As we show in the original analysis, nuclear cycle variations in the boundary IS were low. However, Leiden cluster analysis (3.7) shows that clusters with the lowest IS means reside in the region mostly occupied by nc14 nuclei, while those with the highest IS means reside in the area largely populated by nc11/12 nuclei. Therefore, TAD insulation contributes to the distribution of nuclei in the UMAP.
- Transcription of the doc genes begins at nc14 ⁵, and there is ample direct evidence in the literature linking transcriptional activation and chromatin structure ^{11,14,15}. Therefore, it was sensible to test whether transcriptional status affects the distribution of nuclei within the UMAP. Remarkably, we found that chromatin organization of active and inactive nuclei was indistinguishable at the single nucleus level. Thus, transcriptional activation does not seem to play a role in the distribution of nuclei in the UMAP.

Because of these reasons, we designed an approach that would examine, and directly quantify, the relative contributions of each of these parameters (chromatin condensation, insulation score and transcriptional status) to the distributions of nuclei in the UMAP space. Thus, we believe that the approach is sound and designed to test specific and interesting hypotheses based on previous knowledge. The result that no single parameter can explain the distribution of single nuclei in the UMAP is perhaps not surprising, but it is important to understand the relative contribution of different parameters to chromatin structure.

To address this comment, we revised the text considerably to clearly justify our approach. In addition, we implemented more complex analyses based on Leiden clustering (see 3.7 below). If the reviewer suggests additional analysis, we would be happy to perform them.

This remark was addressed by multiple changes in the text of the revised manuscript:

1. “All in all, we conclude that developmental timing partially explains the distribution of single nuclei in the UMAP space.”

2. “Next, we investigated whether other parameters additionally contributed to this distribution. First, we tested the degree to which TAD condensation influenced the segregation of single nuclei within the UMAP conformational space.... Thus, TAD volume only partially contributes to the global separation of single nuclei between nuclear cycle clusters within the UMAP space.”
3. “As *Drosophila* TADs arise in nc14¹, we reasoned that TAD insulation may determine how single nuclei occupy the UMAP space. To test this hypothesis, we first calculated the ensemble and single nuclei insulation scores (IS) of nc14 nuclei (Fig. 2c, top panel) following the method developed... In summary, only a minority of nuclei exhibit a strong border between TAD1 and doc-TAD in nc14 embryos, and a similarly sized population of insulated TADs is already present at earlier stages of development that do not display discernable ensemble TADs.”
4. “Finally, we applied Leiden clustering to shed further light into the distribution of single nuclei onto the UMAP space (see Methods). All in all, this analysis shows that TAD insulation contributes to the distribution of single nuclei in the UMAP, however, this mapping is not one-to-one and other, multiple hidden structural parameters are likely necessary to describe the position of a single nuclei within the UMAP space.”
5. “The doc genes are specifically expressed at nc14. Thus, we naturally wondered whether their expression contributed to the single-nucleus organization of chromatin at the *doc* locus.... All in all, these results suggest that the chromatin organization of active and inactive nuclei are indistinguishable at the ensemble and single nucleus levels.”

3.3 *We also have a question related to the division of UMAP clusters. In the UMAPs displayed in this paper, closed circular boundaries were drawn around each cluster (nc 11/12 or nc 14). However, in many of the UMAPs, it is hard to see how these boundaries were drawn since the UMAP clusters (for example, nc 11/12 vs. nc 14) were not well separated. Without these boundaries, the underlying distribution seems more intermixed to the eye. Can the authors specify how these cluster boundaries were calculated or defined?*

We apologize for the lack of detail in our original submission, where we have manually demarcated the boundaries of each cluster. To address this comment, we implemented an automatic segmentation method relying on the Gaussian kernel estimate of the difference of density maps (see Methods, section “*Density-based cluster boundaries in UMAP plots*”) and applied it to demarcate the clusters in the UMAPs of Figs. 1g, 1i, 2c, 2g, 3d-e. The results of this algorithm are very similar to our manually drawn profiles, as seen when comparing our original and revised figures (see Table below). In addition, we quantified the proportions of single cells/nuclei within each cluster and added a new panel with this information to Figure 1 (see also Table below).

In summary, we made the following changes to the Figures and text to address this issue:

- The method to automatically quantify clusters is described in the “*Density-based cluster boundaries in UMAP plots*” section of the Methods.
- We updated Figures 1g and 1i, and added the quantifications as two new figure panels (Figs. 1h, 1j).
- We added the following sentences to the text in the Results section:

“We note that a small number of cohesin-depleted single cells (~3%) localized to the region occupied by untreated cells, and vice versa (~8%, Fig. 1h).”

“A small number of nc11/12 single nuclei (~17%) was found in the nc14 cluster, while ~20% of nc14 single nuclei localized to the nc11/12 cluster (Fig. 1j).”

3.4 Related to the previous point: In the discussion associated with Fig 1h: The authors claimed that most of the cells from nc 11/12 and nc 14 segregated into their own UMAP clusters. However, there is certainly some intermixing in the UMAP space: many of the nc

11/12 nuclei, in particular, are present within the boundaries drawn for nc 14. Can the authors provide the exact percentages of how many cells crossed over to the other cluster, like they did for the discussion associated with Fig 1g?

To address this comment, we have used the automatic demarcations (see answer to point 3.3 above) to quantify the exact percentages of nuclei found in each cluster and in the region outside the demarcated areas. The revised manuscript now displays two new panels (Figs. 1h, 1j) with these quantifications, in addition to a revised text (see answer 3.3 above).

3.5 *In the discussion associated with Fig 2a: The authors claimed that averaged distance matrices of the nuclei from the east and west parts of the UMAP (top two in that column) were similar to Fig 1b. However, they looked really different from either the nc 11 or the nc 14 matrix in Fig 1b. and this similarity is really not visible to the eye.*

We apologize for the misunderstanding. The matrices in the original Fig. 2a represented difference matrices, built by subtracting the Hi-M PWD matrix of a specific subset of cells and the Hi-M PWD matrix containing all nuclei (i.e. the ensemble PWD matrix). Thus, matrices that were similar to the ensemble matrix had pixel values close to zero (white in the original Fig. 2a). In contrast, regions of the matrix with higher PWD values than the ensemble appeared in red, and regions with lower PWD values than the ensemble appeared in blue.

In **3.6**, the reviewer argues about the relevance of this analysis where the UMAP is divided into North, South, East, and West, since the axes on the UMAP have no real biological meaning, and can be easily rotated or flipped with some minor changes in the input data or embedding parameters. Indeed, we agree with this overall criticism, and therefore we have removed this analysis from the revised version of the manuscript.

Can the authors provide a correlation number between the distance matrices to demonstrate this similarity? Similarly, it would be great if the authors can provide a correlation number to show that the south and north portions from the UMAP are similar to Fig 1c, as they also claimed.

Following the advice of the reviewer in **3.6**, we have removed the analysis of Fig. 2a from this revision of the manuscript.

3.6 *And, related to the previous point: We are also uncertain whether dividing up the UMAP into North, South, East, and West is truly meaningful, since the axes on the UMAP have no real biological meaning, and can be easily rotated or flipped with some minor changes in the input data or embedding parameters.*

We agree with the referee in that this choice of coordinate system may not be the most appropriate, and that the axes of the UMAP cannot be necessarily decomposed into a simple association of the input parameters, as for other analysis methods like PCA. Thus, this analysis may not be truly meaningful, and therefore we decided to remove it from the revised version of the manuscript.

3.7 Related to the general theme of this section, can the authors come up with some alternative ways of interpreting their UMAPs? For example, can the authors explore Leiden / Louvain clustering on the UMAPs and see if they can retrieve distinct nuclei clusters? How do the Leiden / Louvain clusters overlap with the nc 11/12 vs. nc 14 groups, and how do the structural parameters (radius of gyration, etc.) map onto the Leiden / Louvain clusters and / or the cell cycle groups?

We followed the reviewer's advice and applied Leiden clustering analysis on the UMAPs, and calculated how the structural parameters (radius of gyration/ insulation score) map onto these clusters and cell cycle groups. This analysis shows that Leiden clusters map approximately to the nuclear cycle clusters, however this mapping is not strict (see panel **g** below). Nc11/12 is decomposed mainly into two Leiden clusters (3 and 4), nc14 is decomposed mainly into three Leiden clusters (0, 1, 2), while a final Leiden cluster (5) falls in between the nc14, nc11/12 clusters. The radii of gyration of the Leiden clusters are widely distributed and their means are similar amongst clusters (panel **h** below), consistent with our analysis in the original submission. The insulation scores were also widely distributed, but their means change from cluster to cluster (panel **h** below). In particular, the Leiden cluster with the lowest IS mean (Leiden cluster 1) overlaps widely with the nc14 cluster. This is consistent with the prevalence of low IS values in this region of the UMAP (see Fig. 2f). In contrast, the Leiden clusters with the highest IS means were to a large degree located within the nc11/12 cluster (Leiden cluster 4) or in the region between nc14 and nc11/12 (Leiden cluster 5). This finding is consistent with these regions of the UMAP displaying slightly larger IS values (see Fig. 2f). Finally, Leiden clusters exhibiting low and high IS mean values can also contain a mix of nuclear cycles (Leiden clusters 2 and 5). All in all, these analyses show that nc14 nuclei tend to display lower IS values than nc11/12 nuclei, however, this mapping is not one-to-one and other hidden structural parameters are likely necessary to describe the position of a single nuclei within the UMAP space.

This remark of the reviewer was addressed by adding the two new panels shown above (revised Figs. 2g-h), and by revising the text as follows:

“Finally, we applied Leiden clustering to shed further light into the distribution of single nuclei onto the UMAP space (see Methods). Following Occam’s razor, we used a small number of Leiden clusters (6, Fig. 2g), however, use of more clusters lead to similar conclusions (Figs. S2d-e). Leiden clusters mapped approximately to nuclear cycle clusters, however this mapping was not strict. The nc11/12 cluster was decomposed mainly into two Leiden clusters (3 and 4), while the nc14 cluster was

decomposed mainly into three Leiden clusters (0, 1, 2). A final Leiden cluster (5) fell in between the nc14, nc11/12 clusters. To study the structural differences between Leiden clusters, we mapped the structural parameters used above (radius of gyration and insulation score, Fig. 2h). The radii of gyration were widely distributed and their means were similar amongst clusters, consistent with our previous analysis (Fig. 2a-b). The insulation scores were also widely distributed, but their means displayed more variability. In particular, the Leiden cluster with the lowest IS mean (Leiden cluster 1) overlapped widely with the nc14 cluster. This is consistent with the prevalence of low IS values in this region of the UMAP (Fig. 2f). In contrast, the Leiden clusters with the highest IS means were, to a large degree, located within the nc11/12 cluster (Leiden cluster 4) or in the region between nc14 and nc11/12 (Leiden cluster 5). This finding is consistent with these regions of the UMAP displaying slightly larger IS values (Fig. 2f). Finally, Leiden clusters exhibiting low and high IS mean values also contained a mix of nuclear cycles (Leiden clusters 2 and 5). All in all, this analysis shows that TAD insulation contributes to the distribution of single nuclei in the UMAP, however, this mapping is not one-to-one and other, multiple hidden structural parameters are likely necessary to describe the position of a single nuclei within the UMAP space.”

In addition, the methods section was appended to describe the methods used for Leiden clustering.

3.8 *This is a stand-alone point not related to UMAPs: In the discussion associated with Fig 4d.: the authors picked 0.25 um as the contact threshold, and discovered that the distribution of the number of contacts are similar with or without doc1 activation. Can the authors explain how they selected this threshold, or whether they had experimented with other contact distance thresholds to reach the same conclusion?*

We apologize for the lack of clarity on the selection of the proximity threshold. Following a procedure documented by Su *et al.* ¹¹, we selected the distance threshold by cross-correlating the Hi-M proximity matrices for different distance thresholds with the Hi-C contact matrix of the same specimen (nc14 embryos). The results of this analysis, when compared to the nc14 late HiC dataset of Ogiyama *et al.* ²⁰, are shown in the panel below. This panel has now been included as Fig. S4a in the revised manuscript.

This figure shows that thresholds between 100-300 nm result in very similar correlation scores. As for higher thresholds we get better statistics, we selected 250 nm as our proximity threshold. This explanation is now provided in the legend of Fig. S4a.

3.9 *Another concern here is whether the selection of barcodes is too sparse to capture enhancer-promoter interactions between the two TAD regions*

In Fig. 1b, we displayed the genomic coordinates of our barcodes together with the genes in this region and with ATACseq, and RNAP2 profiles. The barcodes in our design cover the major ATAC peaks in TAD1 (barcodes 1, 8, 11, 13) and in the doc-TAD (barcodes 14, 15, 17, 18), as well as most of the RNAP2 bound regions within these two TADs (barcodes 1, 11, 13, 14, 15, 16-17, 20). Thus, we conclude that the majority of the enhancers and promoters are covered by barcodes. Thus, we do not believe that genomic coverage could account for the weak difference observed in Fig. 4d.

Clarity

For clarity, we have some comments about the interpretation of the figures and disambiguation of the text.

3.10 *In the discussion associated with Fig 1c: The authors claimed that the most notable difference between the two different nuclear cycles is that the distance between TAD1 and doc-TAD barcodes increased in nc14. However, it is difficult to see this overall trend within the pink box in Fig 1c, as there seem to be as many dark pink squares (increases) as there are dark blue squares (decreases) in the pink box. Can the authors clarify what they mean, or find alternative ways to visualize these data to support their claim?*

We agree with the reviewer in that this overall trend is difficult to extract from the pink box in Fig. 1c, as many distances are higher in nc14 than in nc11/12 but some are smaller. To make a more quantitative estimation, in the figure below we plotted PWD distance changes between nc14 and nc11/12 (i.e. bins in Fig. 1c) for: (1) bins within the yellow box, (2) bins within the pink box, and (3) all the bins in the matrix (black).

This analysis clearly shows that distances in the yellow region are higher in nc11/12 than in nc14 embryos. However, it is also apparent that the pink box contains bins with positive and negative distance differences, as pointed out by the reviewer. Thus, despite an overall compaction of the locus (Fig. S2b), a considerable subset of distances between TAD1 and doc-TAD barcodes increase during nc14. In view of these results, we have revised the text as follows:

“The most notable difference between nc11/12 and nc14 resided in an overall decrease in distances within TAD1 (Fig. 1c, yellow box), and an overall condensation of doc-TAD in nc14 nuclei (Fig. S1e). Notably, a considerable number of pairwise distances between TAD1 and doc-TAD barcodes (Fig. 1c, pink box) increased in nc14, despite the overall compaction of the locus in this nuclear cycle (see below).“

3.11 *Under the section that begins with "Single nuclei displaying insulated TADs are common ...", last paragraph, last sentence, the authors said that "this population of insulated TAD is already present at earlier stages of development ..." Were the authors trying to make the point that the nuclei that are insulated in nc14 are the same ones that are already insulated in nc 11/12 and then carried over to nc 14, or simply the percentages are similar in both nuclear cycles? The authors might want to modify the statement to disambiguate*

We apologize for the poor phrasing. To disambiguate, we revised the text as follows:

“In summary, only a minority of nuclei exhibit a strong border between TAD1 and doc-TAD in nc14 embryos, and a similarly-sized population of insulated TADs is already present at earlier stages of development.”

3.12 *We find the last paragraph in the Discussions sections, the conclusion of this paper, too speculative and not fully supported by the data shown. In particular, we are confused by the statement "flexible chromatin organization ... that can extend beyond the domain borders", since it seems to contradict the discussion associated with Fig 4: that the interaction between TAD1 and doc-TAD is not affected by doc1 activation, and therefore showing no evidence of enhancer-promoter interactions between neighboring TADs. Can the authors clarify what they mean or disambiguate?*

We agree with the reviewer in that our data shows no evidence for EP interactions between neighboring TADs. In the sentence cited by the reviewer in our original manuscript, we did not intend to make a specific conclusion about inter-TAD EP interactions contributing to gene regulation. Therefore, we revised the sentence as follows:

“Overall, our analysis of single-nucleus microscopy-based chromosome conformation capture data is compatible with a model of flexible chromatin organization within TADs that serves as a scaffold with enhancer-bound transcription-activating factors encoding the logic that integrates multiple, potentially short-lived, interactions that can extend beyond domain borders defined from ensemble-averaged experiments.“

Minor Concerns

3.13 *This paper focused on the structure of only one chromatin locus. However, the title, "Multiple parameters shape the 3D chromatin structure of single nuclei", suggests a general conclusion. We are concerned that the discoveries from a single chromatin locus might not be representative of the entire nucleus, and suggest that the authors modify the title*

To address this concern, we modified the title to:

Multiple parameters shape the 3D chromatin structure of single nuclei at the doc locus in Drosophila

3.14 *Fig 1b, 1c: The font on the color bar is too small and unlegible, making it hard to tell which color means higher distance.*

We changed the font sizes to make them legible.

3.15 *Fig 3b: The distance maps lack a color bar. In addition, the authors should specify that by "intra-TAD", they mean doc-TAD not TAD1*

We have indicated that intra-TAD means doc-TAD in the figure legend as follows:

"...Intra-TAD distances were calculated for the doc-TAD only, as shown on the region highlighted above."

3.16 *Fig S1f: The authors should specify which color in the UMAP means cohesin-treated, and which one is untreated*

We added legends for cohesin-treated and untreated as well for nc14 and nc11/12 in Figs. S1f-g.

3.17 *We found the font size to be too small for some figure legends and axes*

We increased the font sizes in Figs. 1b, 1c, 2d, and 2d.

References

1. Hug, C. B., Grimaldi, A. G., Kruse, K. & Vaquerizas, J. M. Chromatin Architecture Emerges during Zygotic Genome Activation Independent of Transcription. *Cell* **169**, 216–228.e19 (2017).
2. Dorsett, D. The Many Roles of Cohesin in Drosophila Gene Transcription. *Trends Genet.* **35**, 542–551 (2019).
3. Ramírez, F., Bhardwaj, V., Arrigoni, L., Lam, K. C., Grüning, B. A., Villaveces, J., Habermann, B., Akhtar, A. & Manke, T. High-resolution TADs reveal DNA sequences underlying genome organization in flies. *Nat. Commun.* **9**, 189 (2018).
4. Reim, I., Lee, H.-H. & Frasch, M. The T-box-encoding Dorsocross genes function in amnioserosa development and the patterning of the dorsolateral germ band downstream of Dpp. *Development* **130**, 3187–3204 (2003).
5. Espinola, S. M., Götz, M., Bellec, M., Messina, O., Fiche, J.-B., Houbron, C., Dejean, M., Reim, I., Cardozo Gizzi, A. M., Lagha, M. & Nollmann, M. Cis-regulatory chromatin loops arise before TADs and gene activation, and are independent of cell fate during early Drosophila development. *Nat. Genet.* **53**, 477–486 (2021).
6. Reim, I. & Frasch, M. The Dorsocross T-box genes are key components of the regulatory network controlling early cardiogenesis in Drosophila. *Development* **132**, 4911–4925 (2005).
7. Frankel, N., Davis, G. K., Vargas, D., Wang, S., Payre, F. & Stern, D. L. Phenotypic robustness conferred by apparently redundant transcriptional enhancers. *Nature* **466**, 490–493 (2010).
8. Perry, M. W., Boettiger, A. N. & Levine, M. Multiple enhancers ensure precision of gap gene-expression patterns in the Drosophila embryo. *Proc. Natl. Acad. Sci. U. S. A.* **108**, 13570–13575 (2011).
9. Osterwalder, M., Barozzi, I., Tissières, V., Fukuda-Yuzawa, Y., Mannion, B. J., Afzal, S. Y., Lee, E. A., Zhu, Y., Plajzer-Frick, I., Pickle, C. S., Kato, M., Garvin, T. H., Pham, Q.

- T., Harrington, A. N., Akiyama, J. A., Afzal, V., Lopez-Rios, J., Dickel, D. E., Visel, A. & Pennacchio, L. A. Enhancer redundancy provides phenotypic robustness in mammalian development. *Nature* **554**, 239–243 (2018).
10. Boettiger, A. N., Bintu, B., Moffitt, J. R., Wang, S., Beliveau, B. J., Fudenberg, G., Imakaev, M., Mirny, L. A., Wu, C.-T. & Zhuang, X. Super-resolution imaging reveals distinct chromatin folding for different epigenetic states. *Nature* (2016).
doi:10.1038/nature16496
 11. Su, J.-H., Zheng, P., Kinrot, S. S., Bintu, B. & Zhuang, X. Genome-Scale Imaging of the 3D Organization and Transcriptional Activity of Chromatin. *Cell* **182**, 1641–1659.e26 (2020).
 12. Martin, R. M. & Cardoso, M. C. Chromatin condensation modulates access and binding of nuclear proteins. *FASEB J.* **24**, 1066–1072 (2010).
 13. Ishihara, S., Sasagawa, Y., Kameda, T., Yamashita, H., Umeda, M., Kotomura, N., Abe, M., Shimono, Y. & Nikaido, I. Local states of chromatin compaction at transcription start sites control transcription levels. *Nucleic Acids Res.* **49**, 8007–8023 (2021).
 14. Cardozo Gizzi, A. M., Cattoni, D. I., Fiche, J.-B., Espinola, S. M., Gurgo, J., Messina, O., Houbron, C., Ogiyama, Y., Papadopoulos, G. L., Cavalli, G., Lagha, M. & Nollmann, M. Microscopy-Based Chromosome Conformation Capture Enables Simultaneous Visualization of Genome Organization and Transcription in Intact Organisms. *Mol. Cell* (2019). doi:10.1016/j.molcel.2019.01.011
 15. Mateo, L. J., Murphy, S. E., Hafner, A., Cinquini, I. S., Walker, C. A. & Boettiger, A. N. Visualizing DNA folding and RNA in embryos at single-cell resolution. *Nature* **568**, 49–54 (2019).
 16. Naumova, N., Imakaev, M., Fudenberg, G., Zhan, Y., Lajoie, B. R., Mirny, L. A. & Dekker, J. Organization of the mitotic chromosome. *Science* **342**, 948–953 (2013).
 17. Ma, Y., Kanakousaki, K. & Buttitta, L. How the cell cycle impacts chromatin architecture and influences cell fate. *Front. Genet.* **6**, 19 (2015).
 18. Belmont, A. & Nicolini, C. The G1 period. Two cycles of chromatin conformational

- changes monitored by single cell dye intercalation. *Cell Biophys.* **5**, (1983).
19. Blythe, S. A. & Wieschaus, E. F. Establishment and maintenance of heritable chromatin structure during early embryogenesis. *Elife* **5**, (2016).
 20. Ogiyama, Y., Schuettengruber, B., Papadopoulos, G. L., Chang, J.-M. & Cavalli, G. Polycomb-Dependent Chromatin Looping Contributes to Gene Silencing during *Drosophila* Development. *Mol. Cell* **71**, 73–88.e5 (2018).

REVIEWERS' COMMENTS

Reviewer #3 (Remarks to the Author):

The authors have carefully responded to our comments and we believe the manuscript is now in a good shape for publication.